# Bone marrow endosteal stem cells dictate active osteogenesis and aggressive tumorigenesis

Yuki Matsushita[1,2,6], Jialin Liu [ ][3,6], Angel Ka Yan Chu[3], Chiaki Tsutsumi-Arai[1], Mizuki Nagata[1], Yuki Arai[1], Wanida Ono[1], Kouhei Yamamoto[4], Thomas L. Saunders [ ][5], Joshua D. Welch[3] ✉ & Noriaki Ono [ ][1] ✉

The bone marrow contains various populations of skeletal stem cells (SSCs) in the stromal compartment, which are important regulators of bone formation. It is well-described that leptin receptor (LepR)[+] perivascular stromal cells provide a major source of bone-forming osteoblasts in adult and aged bone marrow. However, the identity of SSCs in young bone marrow and how they coordinate active bone formation remains unclear. Here we show that bone marrow endosteal SSCs are defined by *fibroblast growth factor receptor 3* (*Fgfr3*) and osteoblast-chondrocyte transitional (OCT) identities with some characteristics of bone osteoblasts and chondrocytes. These *Fgfr3-creER-* marked endosteal stromal cells contribute to a stem cell fraction in young stages, which is later replaced by *Lepr-cre*-marked stromal cells in adult stages. Further, Fgfr3[+] endosteal stromal cells give rise to aggressive osteosarcoma- like lesions upon loss of p53 tumor suppressor through unregulated self- renewal and aberrant osteogenic fates. Therefore, Fgfr3[+] endosteal SSCs are abundant in young bone marrow and provide a robust source of osteoblasts, contributing to both normal and aberrant osteogenesis.

Skeletal stem cells (SSCs) are a bone-specific subtype of somatic stem cells with two hallmark capabilities of self-renewal and multipotency[1]. SSCs provide a perpetual source of osteoblasts, chondrocytes, marrow stromal cells, and adipocytes, therefore, play important roles in bone growth and regeneration[2]. SSCs represent compartment-specific stem cells with varied differentiation potential, residing in bone marrow[3–7], periosteum[8–11], and growth plate[12–14]. In the bone marrow, SSCs have been identified among bone marrow stromal cells (BMSCs), which are heterogeneous cell populations that occupy various positions relative to bone surfaces and marrow vasculatures, as revealed by single-cell and spatial transcriptomic studies[15–19]. The two well-described sources of bone-forming osteoblasts are leptin receptor (LepR)-expressing stromal cells in perisinusoidal space in adults[20,21], and hypertrophic

chondrocytes in the growth plate in early life[22,23]. In the adult bone marrow, BMSCs co-expressing LepR and C-X-C motif chemokine ligand 12 (CXCL12), termed CXCL12[+]LepR[+] BMSCs, provide a major source of osteoblasts by including SSCs and other precursor cells[20,21,24], whereas aged SSCs have a decreased bone- and cartilage-forming potential but produce more pre-adipocyte-like stromal cells[25]. In early life, during active bone growth, chondrocyte-to-osteoblast transformation con- tinuously occurring in the vicinity of the growth plate accounts for at least part of the output to osteoblasts[22,26]. However, the identity of bone marrow SSCs during the transition, and how they contribute to active bone formation occurring in young stages remains unclear.

In this study, we report a novel population of SSCs with osteoblast-chondrocyte transitional (OCT) identities in the bone

[1]University of Texas Health Science Center at Houston School of Dentistry, Houston, TX, USA. [2]Department of Cell Biology, Nagasaki University Graduate School of Biomedical Sciences, Nagasaki, Japan. [3]Department of Computational Medicine and Bioinformatics, University of Michigan, Ann Arbor, MI, USA. [4]Department of Comprehensive Pathology, Tokyo Medical and Dental University, Tokyo, Japan. [5]Transgenic Animal Model Core, University of Michigan Medical School, Ann Arbor, MI, USA. [6]These authors contributed equally: Yuki Matsushita, Jialin Liu. ✉e-mail: welchjd@med.umich.edu; noriaki.ono@uth.tmc.edu

marrow endosteal space, which highly expresses *Fgfr3* and contributes to both normal and aberrant osteogenesis. These Fgfr3[+] stem/stromal cells with OCT identities are abundant in the young bone marrow and depleted in the old bone marrow, denoting their transitional nature. Of note, we define OCT identities as a state with some characteristics of both osteoblasts and chondrocytes, instead of cell-type plasticity between osteoblasts and chondrocytes. The term "transitional" emphasizes the unique feature of these cells that are particularly abundant in the young bone marrow. The discovery of this new class of SSCs is based on the conjunction of evidence from unbiased single-cell molecular profiling and functional dissection of the BMSC lineage hierarchy using in vivo cell lineage analysis. Overall, our findings indicate that Fgfr3[+] endosteal stem/stromal cells with OCT identities dictate active and aggressive osteogenesis, identifying these cells as an important regulator of long-term bone homeostasis.

## Results

### Osteoblast-chondrocyte transitional (OCT) stem cells in young bone marrow

First, to define the unique molecular identities of putative skeletal stem cell populations in young bone marrow, we performed single-cell RNA-seq analysis of bone marrow cells marked by *Prrx1-cre*, which uniformly marks all the skeletal lineage cells in the appendicular skeleton (Supplementary Fig. 1a)[27,28], at P21 (Young) and 18 months (18M, Old) using fluorescence-activated cell sorting (FACS) and the 10X Chromium Single-Cell Gene Expression platform (Supplementary Fig. 1b). The P21 and 18M scRNA-seq datasets were computationally integrated with LIGER[29] (Fig. 1a, b).

As expected, *Prrx1-cre*-marked tdTomato[+] bone marrow stromal cells (BMSCs) encompassed large groups of osteoblastic cells (Osteo) and pre-adipocyte-like reticular cells (Adipo), as well as separate clusters of chondrocytes (Chondro) (Fig. 1c); the existence of "chondrocytes" in BMSC cell preparation in our dataset is consistent with previous studies[16,19]. We also observed a cluster of cells with osteoblast-chondrocyte transitional (OCT) identities positioned in the middle of osteoblast and chondrocyte clusters (Fig. 1c, red dotted line), as well as cells with osteoblast-reticular transitional (ORT) identities in the middle of osteoblast and reticular cell clusters. Interestingly, the "OCT stem" cells demonstrated a transitional state simultaneously expressing molecular markers of chondrocytes (*Acan*), osteoblasts (*Col1a1*), and reticular cells (*Cxcl12*) at a lower level than that of the canonical cell types (Fig. 1d and see Supplementary Fig. 1c for additional markers). These OCT stem cells were abundant at P21, but markedly depleted at 18M (Fig. 1e, center and right panels); this shift in cell populations at 18M was associated with a substantial increase in pre-adipocyte-like reticular cells, as reported previously in ref. 19. The expression of these osteoblast and chondrocyte markers by OCT stem cells was high at P21 but diminished at 18M; at 18M, these cells instead expressed adult skeletal stem cell markers such as *Lepr* and *Ly6a* (encoding Sca1) (Supplementary Fig. 1d), indicating a shift of cell identities between young and adult stages. Computational inference of the sex of each cell revealed that the female and male cells maintained a rather consistent 3:1 ratio across all clusters at P21 (Supplementary Fig. 1e), suggesting that the key clusters we identified were not sex-dependent, and sex differences did not confound our analyses. In the following section, we define OCT identities as a state with some characteristics of both osteoblasts and chondrocytes, but not as a state in which cells are transitioning between osteoblasts and chondrocytes. The term "transitional" emphasizes the unique feature of these cells that are abundant in young bone marrow but depleted in the old bone marrow.

RNA velocity analysis[30,31] revealed that OCT stem cells were predicted to provide a robust cellular origin of osteoblasts, pre-adipocyte-like reticular cells, and their intermediate-state cells (Fig. 1e, left panel, red dotted line). Tracing these velocity vectors backward using

CellRank[32] inferred two putative points of cellular origin at P21, one in OCT stem cells (Fig. 1e, center panel, red dotted line) and the other in resting-zone chondrocytes (Fig. 1e, red arrowhead), a known cellular origin of growth plate chondrocytes[13]. Interestingly, this analysis suggests a shift of cellular origins between young and old stages, with pre-adipocyte-like reticular cells at 18M showing a much higher probability of being a point of origin (Fig. 1e, right panel, blue arrowhead and Supplementary Fig. 1f).

Subsequently, we interrogated the single-cell epigenomic states of these cells using single-nucleus ATAC-seq analysis of the same biological sample as the scRNA-seq dataset using the 10X Genomics ATAC platform. The snATAC-seq dataset was computationally integrated with the isogenic scRNA-seq dataset using LIGER (Fig. 1f). We projected the scRNA and snATAC profiles onto a three-dimensional simplex, with each vertex representing one of the three main classes of bone marrow stromal cells (chondrocytes, osteoblasts, and pre-adipocyte-like reticular cells). We summarized each cell in terms of its transcriptomic or epigenomic similarity to chondrocytes, osteoblasts, and reticular cells (Fig. 1g and Supplementary Fig. 2a). Cells near the middle of the simplex have almost equal similarity to each fate and thus represent relatively unprimed and multipotent cell states (Fig. 1g, for OCT stem cells), while cells near the vertices have a strong similarity to only a single fate and are thus more committed and differentiated (Fig. 1g, lower panels). We defined the vertices of the simplex using the transcriptomic and epigenomic signatures of the most distinct clusters for each fate (Osteoblast 1, Chondrocyte 1, and Reticular 1), based on known lineage markers and UMAP and RNA velocity analyses. In addition to calculating the similarity based on the current gene expression state, we also estimated the future differentiation potential toward each of the three fates using the vectors from RNA velocity. We visualized this potential using three arrows for each position in the simplex, with the relative lengths indicating the strength of the differentiation potential toward each distinct fate (Fig. 1g and Supplementary Fig. 2a). We selected the top 30 differentially expressed genes (DEGs) or differentially accessible peaks (DAPs) from each cluster; the ternary plots showed a very similar trend using 500 or 1,000 differentially expressed features (Supplementary Fig. 3a). Including the whole cell type as the reference (instead of Osteoblast 1, Chondrocyte 1, and Reticular 1) distorted the shape of each plot, demonstrating the negative effect of defining the cell fates using the less pure and more transitional clusters (Supplementary Fig. 3b).

The simplex and velocity analyses clarified the relationships among each of the clusters. Osteoblast 1 and 2 possessed similarity to and potential toward primarily the osteoblast fate; Chondrocyte 1 was most similar to and biased toward the chondrocyte fate; and Reticular 1 and 2 toward the reticular fate (Fig. 1g and Supplementary Fig. 2a). In contrast, cells in the OCT stem clusters demonstrated "trilineage" potential toward all three fates, predominantly toward osteoblast and reticular fates. Intriguingly, the chromatin accessibility profiles of OCT stem cells indicate trilineage potential consistent with the transcriptomic profiles, but with a greater bias toward the reticular fate than predicted from gene expression (Fig. 1g, for OCT stem cells). One possible interpretation of these results is that the epigenomic regulatory landscape of these stromal cells is biased toward the reticular lineage, and cells require some external signals to activate the osteoblast fate. In addition, the chondrocyte fate may be passive and constantly exhausted at the transcription level.

To further validate our observations, we performed joint snRNA and snATAC profiling of *Prrx1-cre*-marked cells using the 10X Multiome platform (Supplementary Fig. 2b). We used LIGER to annotate the snRNA from the Multiome data with the cluster labels from the *Prrx1-cre* scRNA data. We then performed the same simplex and velocity analyses using the multiome data. This allowed us to experimentally validate the computationally inferred relationships between transcriptome and epigenome features. The results confirm our

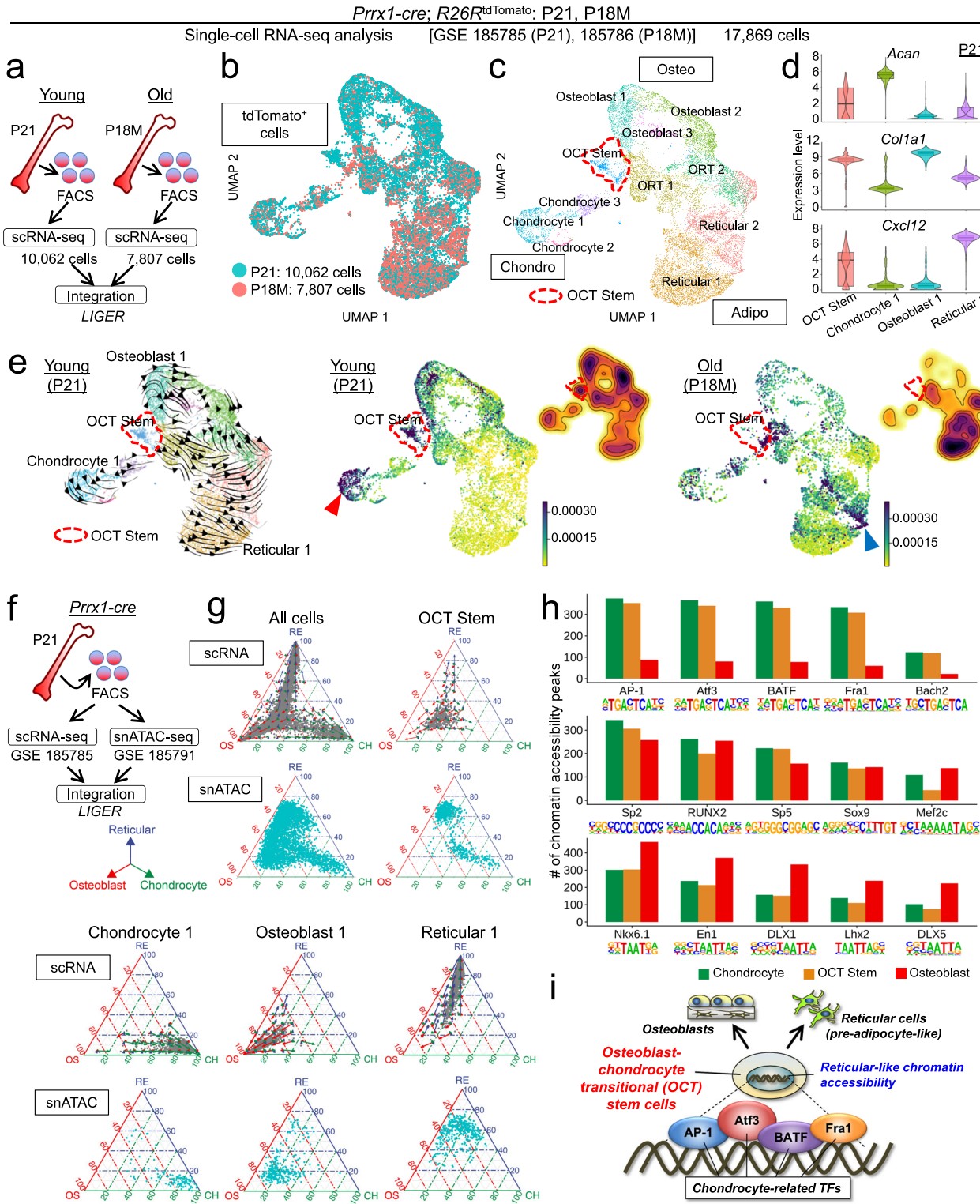

observations that both transcriptome and epigenome profiles of OCT stem clusters indicate trilineage potential (Supplementary Fig. 2b–d).

We also investigated the transcription factors (TFs) predicted to regulate OCT stem cells. TF binding motif enrichment analysis revealed that chromatin accessibility peaks in OCT stem cells are enriched for chondrocyte-related TF binding motifs such as those of AP-1 and ATF3, but have lower levels of accessible osteoblast-related motifs such as DLX1/5 or LHX2 (Fig. 1h). This suggests that the OCT stem cells are still being regulated primarily by chondrocyte-related TFs, supporting their transitional identities. In contrast, osteoblasts,

chondrocytes, and OCT stem cells all show similar levels of binding motifs for TFs such as RUNX family, SP2/5 and SOX9 proteins (Fig. 1h, middle panel).

In summary, these integrative single-cell epigenomic and transcriptomic analyses indicate that putative stem cells with osteoblast-chondrocyte transitional identities are primed for the three types of cells, i.e., osteoblasts, chondrocytes, and pre-adipocyte-like reticular cells, with enrichment of transcription factor binding motifs that typically regulate chondrocyte differentiation (Fig. 1i).

**Fig. 1 | Osteoblast-chondrocyte transitional identities of putative skeletal stem cells in young marrow. a** Diagram for LIGER data integration. Single-cell RNA-seq datasets of sorted tdTomato+ single cells harvested from *Prrx1-cre; R26R^tdTomato* femur bone marrow at P21 (Young, 10,062 cells) and P18M (Old, 7807 cells) were merged by LIGER. **b** UMAP visualization of two datasets (P21 and P18M) merged by LIGER. Cells were pooled from *n* = 3 (P21), *n* = 2 (P18M) mice. Blue: P21 (Young), red: P18M (Old). **c** UMAP visualization of major sub-clusters of Prrx1^cre-tdTomato+ cells, Chondro [Chondrocyte (1–3)], Osteo [Osteoblast (1–3)], Adipo [Reticular (1–2)], osteoblast-reticular transitional (ORT) cells (1–2), and osteoblast-chondrocyte transitional (OCT) stem cells. Red dotted contour: OCT stem cluster. **d** Violin plots of representative chondrocyte (*Acan*), osteoblast (*Col1a1*), and reticular cell (*Cxcl12*) markers in representative clusters (OCT Stem, Chondrocyte 1, Osteoblast 1, and Reticular 1) at P21. *n* = 570 (OCT Stem), *n* = 764 (Chondrocyte 1), *n* = 1181 (Osteoblast 1), and *n* = 1797 (Reticular 1) cells. Data were presented as median (midline in the plot) [25 percentile (lower bound in the plot), 75 percentile (upper bound in the plot)] of the log2-transformed normalized gene expression. **e** RNA velocity at P21 (left), CellRank and density plot at P21 (center), and at P18M (right). Black arrows: dynamic velocity vectors inferring future states. Red dotted contour: OCT Stem cluster. The color scale for CellRank: initial cell state probability. The color scale for density plots: violet: high expression, yellow: low expression. **f** Single-cell RNA-seq and snATAC-seq datasets of fluorescently sorted tdTomato+ single cells harvested from the same biological sample (*Prrx1-cre; R26R^tdTomato* femur bone marrow at P21) were integrated by LIGER. **g** Simplex and velocity analyses of representative cell types. Cells colored in gray show their transcriptomic affinities towards three vertices, whereas the arrows show the future differentiation potential of the cells. Cells colored in aqua show their epigenomic affinities towards three vertices. Blue arrow and axis: Reticular cluster. Red: Osteoblast cluster. Green: Chondrocyte cluster. **h** Barplots of transcription factor (TF) binding sites analysis. Y-axis: number of chromatin accessibility peaks. **i** Proposed model of OCT stem cell. "OCT stem" cells with osteoblast-chondrocyte transitional identities have reticular-like chromatin accessibility, future differentiation potential toward pre-adipocyte-like cells and osteoblasts, and enrichment of TFs that regulate chondrocyte differentiation.

## Fgfr3+ stromal cells are localized to the bone marrow endosteum

The identification of putative skeletal stem cells with osteoblast-chondrocyte transitional identities prompted us to further examine in vivo identities of these cells. First, we searched for potential markers of these OCT stem cells by screening expression patterns of chondrocyte- and osteoblast-related genes based on UMAP plots (Fig. 2a). Interestingly, we found that *fibroblast growth factor receptor 3* (*Fgfr3*) was highly expressed in OCT stem cells (Fig. 2a, red arrow), while *growth arrest-specific 1* (*Gas1*) was highly expressed by ORT cells (Fig. 2a, blue arrow); expression patterns of these genes were distinct from canonical chondrocyte, osteoblast and reticular cell markers such as *Col2a1*, *Col1a1*, and *Lepr*, respectively (Fig. 2a).

*FGFR3* is a causative gene for achondroplasia in humans, and is expressed by cells in bones and cartilages, including proliferating chondrocytes in the growth plate and periosteal cells in the metaphyseal region[33]. To identify an *Fgfr3*-expressing cell type that may correspond to the above-identified OCT stem cells in bone marrow, we first analyzed an *Fgfr3-GFP* bacterial artificial chromosome (BAC) transgenic line (MMRRC:031901). At P21, Fgfr3-GFP+ cells expressed *Fgfr3*, and were localized not only in the growth plate and the distal periosteum, but also in the endosteum, in an identical pattern to FGFR3 proteins (Fig. 2b, left and center panels, arrow). At higher magnification, Fgfr3-GFP+ cells overlaid Osx-mCherry+ cells on the endosteum with many of them co-expressing ALPL (Fig. 2b, right panels). Further, some of the Fgfr3-GFP+ cells in the endosteum also expressed a canonical chondrocyte marker, aggrecan (ACAN) (Fig. 2b, arrowheads in the rightmost panel).

To achieve in vivo cell lineage analyses, we utilized a tamoxifen-inducible *Fgfr3-creER* P1-derived artificial chromosome (PAC) transgenic line[34], and generated a *Gas1-creER* 3'UTR knockin allele by CRISPR/Cas9 (Supplementary Fig. 4a). We performed short-chase analyses of *Fgfr3-creER; R26R^tdTomato* and *Gas1-creER; R26R^tdTomato* mice at P23 (pulsed at P21). First, analysis of *Fgfr3-GFP; Fgfr3-creER; R26R^tdTomato* triple transgenic mice revealed that *Fgfr3-creER*-marked tdTomato+ (hereafter, Fgfr3^CE-tdTomato+) cells were localized to the endosteum with almost all expressing Fgfr3-GFP (98.0 ± 0.2%) (Fig. 2c). Second, we investigated which cellular subsets of BMSCs *Fgfr3-creER* and *Gas1-creER* marks based on their single-cell molecular profiles. To this end, we performed scRNA-seq on FACS-isolated Fgfr3^CE-tdTomato+ and Gas1^CE-tdTomato+ cells at P23 (pulsed at P21) and superimposed these datasets on the Prrx1-cre dataset using LIGER. Fgfr3^CE-tdTomato+ cells encompassed OCT stem cells (red dotted line in Fig. 2d) and chondrocytes, while Gas1^CE-tdTomato+ cells corresponded mainly to ORT cells (blue dotted line in Fig. 2d) and osteoblasts. Therefore, Fgfr3^CE-tdTomato+ cells include OCT stem cells, while Gas1^CE-tdTomato+ cells include transit-amplifying cells. Therefore, we capitalized on FGFR3 as a novel marker of endosteal stromal cells with OCT states in bone marrow in the following analyses.

We further performed flow cytometry analyses. These data revealed that Fgfr3+ cells and Gas1+ cells represented extremely small fractions of non-hematopoietic cells (Fgfr3^CE-tdTomato: 0.026 ± 0.004%, Gas1^CE-tdTomato: 0.031 ± 0.002% of CD45/Ter119/CD31^neg cells), in sharp contrast with Col1a1(2.3 kb)-GFP+, Cxcl12-GFP+ and *LepR-cre*-marked cells that accounted for much larger percentages of CD45/Ter119/CD31^neg cells (Col1a1(2.3 kb)-GFP+: 0.26 ± 0.08%, Cxcl12-GFP+: 0.33 ± 0.02%, LepR^cre-tdTomato+: 0.23 ± 0.07%) (Fig. 2e). Quantitative flow cytometry analysis of mouse SSC (mSSC) markers revealed that a large fraction of Fgfr3^CE-ZsGreen+ cells, as well as Gas1^CE-ZsGreen+ cells expressed mSSC markers, while ~7% of mSSCs were Fgfr3+ (Fig. 2f). In addition, Fgfr3^CE-ZsGreen+ cells contributed to a higher fraction of mSSCs than Gas1^CE-ZsGreen+ cells did (Fig. 2f). Therefore, a significant fraction of Fgfr3+ endosteal stromal cells express mSSC markers.

## Fgfr3+ endosteal stromal cells are enriched for skeletal stem cell activity

We subsequently set out to define whether Fgfr3+ cells possess skeletal stem cell activities. To this end, we performed a colony-forming unit fibroblast (CFU-F) assay of bone marrow and periosteal cells isolated from P23 *Fgfr3-creER; R26R^tdTomato* bones (pulsed at P21) (Fig. 3a and Supplementary Fig. 4b–d). Fgfr3^CE-tdTomato+ cells contributed to a large fraction of CFU-Fs in the bone marrow (76.0 ± 5.8%), while these cells contributed to a much smaller fraction of CFU-Fs in the periosteum (18.1 ± 7.4%) (Fig. 3b, c), indicating that Fgfr3+ bone marrow cells include a large clonogenic population. Comparative analyses of Gas1+ cells using *Gas1-creER*, Gli1+ cells using *Gli1-creER* and LepR+ cells using *Lepr-cre* at the same young stage revealed that these cells contributed to significantly smaller fractions of bone marrow CFU-Fs at this stage (*Gas1-creER*: 37.4 ± 7.2%, *Gli1-creER*: 32.1 ± 4.9%, *LepR-cre*: 3.8 ± 1.8%, Fig. 3b, c). Interestingly, the clonogenic fraction shifted in adult bone marrow from Fgfr3+ cells to LepR-lineage cells (Fig. 3d, e); while Fgfr3+ cells contributed to progressively smaller fractions of bone marrow CFU-Fs (45.8 ± 12.2% at 8 weeks, 20.9 ± 3.8% at 9 months), LepR-lineage cells contributed to a progressively larger fraction of bone marrow CFU-Fs (55.2 ± 9.7% at 8 weeks, 90.7 ± 5.0% at 9 months) (Fig. 3d, e). The contribution of Gas1+ cells at 8 weeks to CFU-Fs was essentially negligible (0.0 ± 0.0%) (Supplementary Fig. 4e). Therefore, Fgfr3+ cells encompass a large majority of CFU-Fs in young −but not in adult− bone marrow.

In addition, a fraction of young bone marrow Fgfr3^CE-tdTomato+ clones possess long-term in vitro passageability; 13.9% (5/36) of Fgfr3^CE-tdTomato+ individual clones could be passaged over eight generations (Fig. 3c). These cells have the capability to

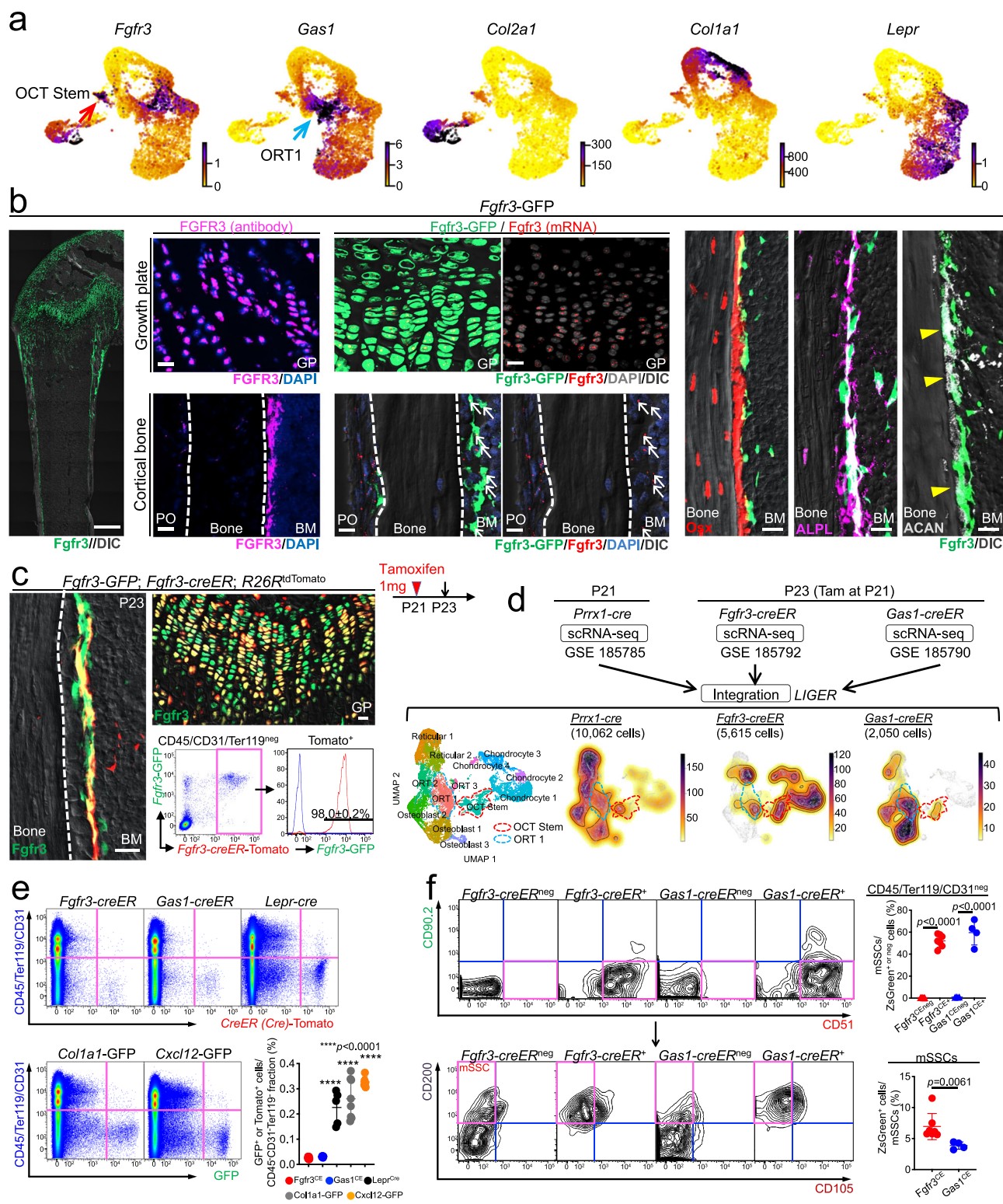

generate Alcian Blue+ spheres, Alizarin Red+ mineralized matrices and Oil red O+/LipidTOX+ lipid droplets with the upregulation of chondrocyte, osteoblast, and adipocyte markers under chondro-genic, osteogenic and adipogenic conditions, respectively (Fig. 3f, g). In contrast, only 2.2% (1/46) of periosteal Fgfr3+ clones survived beyond Passage 4 (Fig. 3c), indicating their transient nature. Therefore, Fgfr3+ cells in the young bone marrow are highly enri-ched for ex vivo colony-forming activities, a fraction of which possess robust in vitro self-renewability and trilineage differ-entiation potential.

## Robust osteogenic capabilities of Fgfr3+ endosteal stromal cells in homeostasis

Subsequently, we defined in vivo localization and cell fates of Fgfr3+ cells in young bone marrow (Fig. 4a). Short-chase analysis of *Fgfr3-creER*; *R26R^tdTomato* mice at P23 (pulsed at P21) demonstrated that Fgfr3CE-tdTomato+ marrow stromal cells in the endosteal space did not apparently overlap with Col1a1(2.3 kb)-GFP+ osteoblasts or Cxcl12-GFP+ reticular cells[35] (Fig. 4b and Supplementary Fig. 5a). Comparative short-chase analysis of *Gas1-creER*; *R26R^tdTomato* revealed that Gas1CE-tdTomato+ cells were also localized to the endosteal space (Fig. 4c and

**Fig. 2 | Fgfr3⁺ stromal cells are localized to the bone marrow endosteum.**
**a** UMAP plots colored by expression of representative OCT Stem (*Fgfr3*), ORT 1 (*Gas1*), chondrocyte (*Col2a1*), osteoblast (*Col1a1*), and reticular cell (*Lepr*) markers. Violet: high expression, yellow: low expression. **b** *Fgfr3-GFP* femur at P21. Left panel: Scale bar: 500 μm. *n* = 4 mice. Center panels: Growth plate (upper) and cortical bone (lower) in high magnification. Immunohistostaining for FGFR3 (left) and RNAscope analyses of *Fgfr3* (right). Scale bar: 20 μm. *n* = 4 mice. Right panels: Magnified images of endosteal space with *Osx-mCherry* (left), ALPL staining (center), and ACAN staining (right). GP: growth plate, PO: periosteum, BM: bone marrow. Scale bar: 20 μm. *n* = 4 mice. **c** *Fgfr3-GFP*; *Fgfr3-creER*; *R26R^tdTomato* femur at P23 (pulsed at P21). Histology and flow cytometry analysis of CD45/Ter119/CD31^neg bone marrow cells isolated from distal femurs. Fgfr3-GFP expression in Fgfr3^CE-tdTomato⁺ cells. Blue lines: GFP^neg control cells. GP: growth plate, BM: bone marrow. Scale bar: 20 μm. *n* = 3 mice. **d** UMAP visualization of three datasets (Prrx1^cre-tdTomato⁺ cells at P21, Fgfr3^CE-tdTomato⁺ cells at P23, and Gas1^CE-tdTomato⁺ cells at P23) merged by LIGER. Prrx1^cre-tdTomato⁺: 10,062 cells, pooled from *n* = 3 mice,

Fgfr3^CE-tdTomato⁺: 5615 cells, pooled from *n* = 8 mice (pulsed at P21), Gas1^CE-tdTomato⁺: 2050 cells, pooled from *n* = 7 mice (pulsed at P21). Leftmost: UMAP visualization of major sub-clusters. Center to the right: Density plots of *Prrx1-cre* (left center), *Fgfr3-creER* (right center), and *Gas1-creER* (rightmost). Red dotted contour: OCT Stem cluster (enriched in *Fgfr3-creER* dataset). Blue dotted contour: ORT 1 cluster (enriched in *Gas1-creER* dataset). Colored scale, violet: high expression, yellow: low expression. **e** Flow cytometry analyses of CD45/Ter119/CD31^neg bone marrow cells isolated from *Fgfr3-creER*; *R26R^tdTomato* and *Gas1-creER*; *R26R^tdTomato* femurs at P23 (pulsed at P21), or *Lepr-cre*; *R26R^tdTomato*, *Col1a1-GFP*, and *Cxcl12-GFP* femurs at P21. Lower right: Percentage of tdTomato⁺ cells within CD45/Ter119/CD31^neg fraction. *n* = 7 (Fgfr3^CE), *n* = 6 (Gas1^CE), *n* = 5 (Lepr^cre), *n* = 8 (Col1a1-GFP), and *n* = 6 (Cxcl12-GFP) mice. Two-tailed, one-way ANOVA followed by Dunnett's multiple comparison test. **f** Percentage of mSSCs among Fgfr3^CE+/neg or Gas1^CE+/neg cells (upper). Percentage of Fgfr3^CE+ and Gas1^CE+ cells among mSSCs (lower). *n* = 7 (Fgfr3^CE), *n* = 4 (Gas1^CE) mice. Data were presented as mean ± s.d. Exact *P* value is indicated in the figures. Source data are provided as a Source Data file.

Supplementary Fig. 5b). Fgfr3^CE-tdTomato⁺ marrow stromal cells did not express GAS1 shortly after the pulse at P23 but expressed GAS1 after 7 days of the chase at P28 (Supplementary Fig. 5a), supporting our model that Fgfr3⁺ cells are precursors of Gas1⁺ cells. Importantly, *Fgfr3-creER* and *Gas1-creER* lines described here show little to no promiscuity in the marrow stromal compartment in the absence of tamoxifen at all stages examined (Supplementary Fig. 5c).

Moreover, only a small fraction of these *creER*-marked cells overlapped with Cxcl12-GFP⁺ cells (Fgfr3^CE-tdTomato: 5.3 ± 0.8%, Gas1^CE-tdTomato: 1.8 ± 0.6% of Cxcl12-GFP⁺ cells, Fig. 4d), in stark contrast with *Lepr-cre* that marked the majority of Cxcl12-GFP⁺ cells (68.3±8.2%, Fig. 4d). Similarly, only a small fraction of these *creER*-marked cells overlapped with Col1a1(2.3 kb)-GFP⁺ cells (Fgfr3^CE-tdTomato: 1.2 ± 0.6%, Gas1^CE-tdTomato: 4.8 ± 1.3% of Col1a1(2.3 kb)-GFP⁺ cells, Fig. 4d, lower panel). These analyses highlight the nature of Fgfr3⁺ and Gas1⁺ cells in the endosteal space that are largely distinct from osteoblasts on the bone surface or Cxcl12⁺LepR⁺ reticular stromal cells in the marrow space.

We further defined the long-term cell fates of Fgfr3⁺ cells in homeostasis. *Fgfr3-creER*; *R26R^tdTomato* or *Gas1-creER*; *R26R^tdTomato* carrying either *Col1(2.3 kb)-GFP* or *Cxcl12^GFP/+* reporter were pulsed at P21 (hereafter, Fgfr3^CE-P21 cells) and chased for a period up to one year (Fig. 4a). Fgfr3^CE-P21 cells expanded substantially within the metaphyseal marrow space after 4 weeks of the chase and contributed to a large number of Col1a1(2.3 kb)-GFP⁺ osteoblasts and osteocytes of the trabecular bone and the endosteal surface (Fig. 4e). Fgfr3^CE-P21 cells continued to contribute to a large number of osteoblasts and marrow reticular stromal cells in the metaphyseal space during the extended period up to 1 year of chase (Fig. 4e and Supplementary Fig. 5d, quantification shown in Fig. 4g, red bars). Likewise, Gas1^CE-P21 cells expanded within the metaphyseal marrow space and contributed to a substantial number of Col1a1(2.3 kb)-GFP⁺ osteoblasts after 4 weeks of chase (Fig. 4f); however, Gas1^CE-P21 cells disappeared initially from the trabecular bone surface after 8 weeks of the chase, and later from the metaphyseal marrow stroma after 6 months of chase (Fig. 4f and quantification shown in 4g, blue bars). We also pulsed *Col1a1*(2.3 kb)-*GFP*; *Fgfr3-creER*; *R26R^tdTomato* mice at 8W (Fgfr3^CE-8W cells). Fgfr3^CE-8W cells were localized in the endosteal space after 2 days of the chase and contributed to osteoblasts and marrow stromal cells after 1 year of the chase, albeit to a lesser extent than Fgfr3^CE-P21 cells did (Supplementary Fig. 5e).

Further, quantitative flow cytometry analyses of Fgfr3^CE-P21 cells revealed that the fraction of Fgfr3^CE-P21 cells among osteoblasts and reticular stromal cells increased progressively as the chase period extended (Col1a1-GFP⁺: from 1.2 ± 0.6% to 63.5 ± 4.9%, Cxcl12-GFP⁺: from 5.3 ± 0.8% to 41.7 ± 2.0% for the second day and the 8th week, respectively, Fig. 4h, red line). In contrast, the fraction of Gas1^CE-P21 cells among osteoblasts and reticular stromal cells initially

increased, but eventually decreased over time (Col1a1-GFP⁺: from 4.8 ± 1.3% to 8.8 ± 3.1%, Cxcl12-GFP⁺: from 1.8 ± 0.6% to 7.4 ± 0.8%, for the second day and the 8th week, respectively, Fig. 4h, blue line). These data support the concept that Fgfr3⁺ cells provide a long-term cellular source of osteoblasts and reticular stromal cells, while Gas1⁺ cells serve only as a transient cellular source of these cells.

To determine how Fgfr3⁺ cells differ from previously described Gli1⁺[36,37], Col2a1⁺[12], or Grem1⁺ cells[38], we analyzed the expression profiles of these genes across cell types within the *Prrx1-cre* (P21) sample. *Fgfr3* showed a different expression profile from *Gli1*, *Col2a1*, and *Grem1* (Supplementary Fig. 6a). Further, we took advantage of *Gli1-creER*; *R26R^tdTomato* and *Col2a1-creER*; *R26R^tdTomato* mice carrying Col1a1(2.3 kb)-GFP reporter, pulsed at P21 (Gli1^CE-P21 and Col2a1^CE-P21 cells, respectively). Shortly after the pulse, Gli1^CE-P21 cells were localized to the growth plate and its adjacent metaphyseal space, whereas Col2a1^CE-P21 cells were almost exclusively localized within the growth plate; importantly, neither Gli1^CE-P21 or Col2a1^CE-P21 cells were localized in the endosteal marrow space at this stage (Supplementary Fig. 6b, c). After the chase, both Gli1^CE-P21 and Col2a1^CE-P21 cells contributed to osteoblasts and marrow stromal cells of the metaphyseal space, as previously reported (Supplementary Fig. 6b, c)[12,36]. However, their contribution was limited to the proximity to the growth plate, while Fgfr3^CE-P21 cells contributed to a far-reaching domain of the marrow space (quantification shown in Supplementary Fig. 6d, red bars, versus Gli1^CE-P21 cells in orange bars and Col2a1^CE-P21 in purple bars). *Gli1-creER*; *R26R^tdTomato* and *Col2a1-creER*; *R26R^tdTomato* mice showed tamoxifen-independent labeling of osteoblasts and osteocytes in the cortical bone (Supplementary Fig. 6e). In addition, Fgfr3⁺ cells did not express GREM1 in the endosteum[38] (Supplementary Fig. 6f). Therefore, *Fgfr3-creER* appears to mark a population of stem cells in the endosteal space that does not coincide with Gli1⁺, Col2a1⁺, or Grem1⁺ cells in the growth plate, the metaphysis or the endosteum.

## Injury responses of Fgfr3⁺ endosteal stromal cells

We aimed to define the contribution of Fgfr3⁺ endosteal stromal cells to regenerative osteogenesis (Fig. 5a). For this purpose, we first utilized a drill-hole cortical injury model that disrupts the endosteal surface to induce direct recruitment of BMSCs to the injury site (Fig. 5b)[39]. *Fgfr3-creER*; *R26R^tdTomato* and *Gas1-creER*; *R26R^tdTomato* mice were pulsed at P21 and underwent surgery after 7 days of the chase at P28 (Fig. 5b). After 2 days of injury, Fgfr3^CE-P21 cells invaded into the bone defect from the endosteal side, but not from the periosteal side, demonstrating that the drill-hole injury model can specifically recruit Fgfr3⁺ endosteal stromal cells, but not periosteal cells. After 7 days of injury at P35, Fgfr3^CE-P21 cells expanded drastically within the injury site and dominated the space between the intact cortical bones, while Gas1^CE-P21 cells contributed to the injury site to a much lesser extent (Fig. 5b). After 8 weeks of injury, Fgfr3^CE-P21 cells

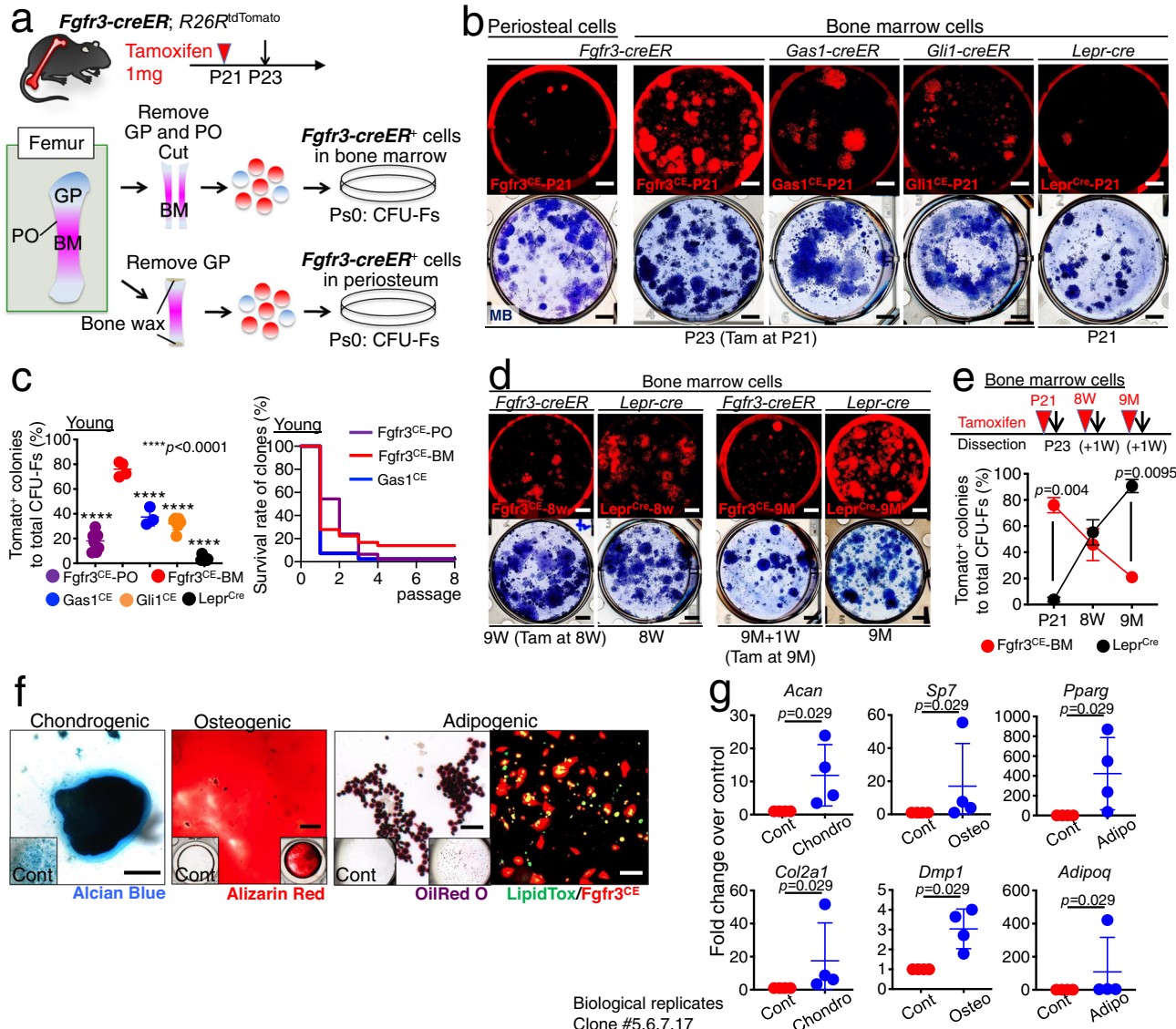

**Fig. 3 | Fgfr3⁺ endosteal stromal cells are enriched for skeletal stem cell activity.** **a** Diagram of protocols to isolate *Fgfr3-creER*⁺ cells from bone marrow (top) or periosteum (bottom). **b** CFU-F assay. Fgfr3^(CE)-tdTomato⁺ periosteal cells (leftmost), Fgfr3^(CE)-tdTomato⁺ (left center), Gas1^(CE)-tdTomato⁺ (center), and Gli1^(CE)-tdTomato⁺ (right center) bone marrow cells at P23 (pulsed at P21), and Lepr^(cre)-tdTomato⁺ (rightmost) bone marrow cells at P21. Upper: tdTomato⁺ colonies, Lower: total colonies stained by methylene blue. Scale bar: 5 mm. **c** Percentage of tdTomato⁺ colonies among total CFU-Fs in young bone (left). Fgfr3^(CE)-periosteum, Fgfr3^(CE)-bone marrow, Gas1^(CE), Gli1^(CE) and Lepr^(cre). $n = 8$ (Fgfr3^(CE)-periosteum), $n = 4$ (Fgfr3^(CE)-bone marrow), $n = 3$ (Gas1^(CE)), $n = 7$ (Gli1^(CE)), $n = 8$ (Lepr^(cre)) mice. Two-tailed, one-way ANOVA followed by Dunnett's multiple comparison test. Survival curve of individual tdTomato⁺ clones over serial passages (right). $n = 46$ (Fgfr3^(CE)-periosteum), $n = 25$ (Fgfr3^(CE)-bone marrow), $n = 41$ (Gas1^(CE)) clones. Log-rank (Mantel–Cox) test. **d** CFU-F assay of Fgfr3^(CE)-tdTomato⁺ bone marrow cells at 1 week after the pulse at 8W or 9M, and Lepr^(cre)-tdTomato⁺ bone marrow cells at 8W and 9M. Scale bar: 5 mm. **e** Percentage of tdTomato⁺ colonies among total CFU-Fs across young and adult stages. Fgfr3^(CE) (pulsed at P21, 8W, or 9M) or Lepr^(cre) bone marrow cells. Fgfr3^(CE): $n = 4$ (P21), $n = 4$ (8W), $n = 6$ (9M). Lepr^(cre): $n = 8$ (P21), $n = 5$ (8W), $n = 4$ (9M). Two-tailed, Mann–Whitney's $U$-test. **f** In vitro trilineage differentiation assay of Fgfr3^(CE)-tdTomato⁺ clones (Passage 2–7). Left: Chondrogenic condition, Alcian Blue staining. Center: Osteogenic conditions, Alizarin Red staining. Right two panels: Adipogenic conditions, Oil red O (right center) and LipidTOX staining (rightmost), Green: LipidTOX-Alexa488, red: tdTomato. Scale bar: 200 μm. **g** qPCR analyses of Fgfr3^(CE)-tdTomato⁺ clones, expression of chondrocyte markers (*Acan*, *Col2a1*), osteoblast markers (*Sp7*, *Dmp1*), adipocyte markers (*Pparg*, *Adipoq*) under chondrogenic, osteogenic and adipogenic conditions, respectively. $n = 4$ clones (#5, 6, 7, 17). Two-tailed, Mann–Whitney's $U$-test. Data were presented as mean ± s.d. Exact $P$ value is indicated in the figures. Source data are provided as a Source Data file.

robustly contributed to a large fraction of osteocytes in the regenerated portion of the cortical bone, while Gas1^(CE)-P21 cells contributed to significantly smaller fractions of cells in the callus and the regenerated cortical bone (tdTomato⁺ osteocytes in the regenerated cortical bone: Fgfr3^(CE)-P21: 58.2 ± 8.0%, Gas1^(CE)-P21: 13.7 ± 2.3%, Fig. 5b), demonstrating that Fgfr3⁺ endosteal stromal cells, but not Gas1⁺ cells, provided an important source of reparative cortical osteoblasts in young bones.

To further define the functional significance of Fgfr3⁺ endosteal stromal cells, we conditionally inactivated Wnt/β-catenin signaling in Fgfr3⁺ cells during cortical bone regeneration. We pulsed littermates of *Fgfr3-creER; Ctnnb1^(fl/+); R26R^(tdTomato)* (Fgfr3-βCat Control) and *Fgfr3-creER; Ctnnb1^(fl/fl); R26R^(tdTomato)* (Fgfr3-βCat cKO) as well as *Gas1-creER; Ctnnb1^(fl/+); R26R^(tdTomato)* (Gas1-βCat Control) and *Gas1-creER; Ctnnb1^(fl/fl); R26R^(tdTomato)* (Gas1-βCat cKO) at P21, performed the drill-hole surgery after 7 days of the chase at P28 and analyzed the defect

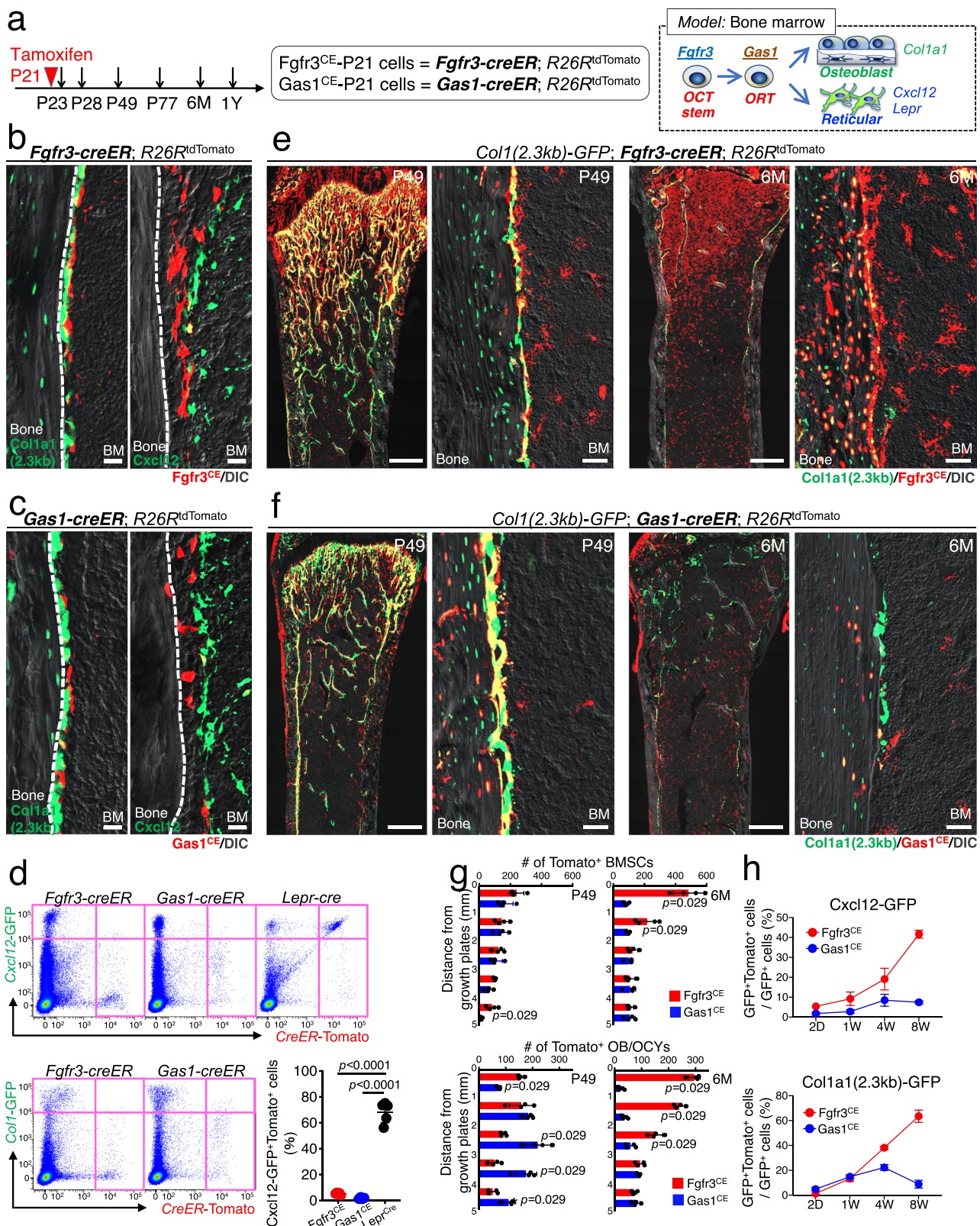

after 14 days of surgery at P42 (Fig. 5c). Bone volume and bone mineral density of the injured cortical area were significantly reduced in Fgfr3-βCat cKO mice, but not in Gas1-βCat cKO mice, compared to Control, associated with a reduction in Col1a1(2.3 kb)-GFP⁺Fgfr3^CE-P21 cells within the defect (Fig. 5c). Therefore, the regenerative potential of Fgfr3⁺ endosteal stromal cells is at least in part regulated by Wnt/β-catenin signaling.

We further performed additional experiments to rule out the contribution of Fgfr3⁺ periosteal cells to regenerative osteogenesis. To achieve this goal, we first performed a periosteal injury surgery by creating a crater on the cortical bone surface, to specifically induce periosteal regenerative responses (Fig. 5d). After 7 days of injury, no Fgfr3^CE-P21 cells were presented within the crater defect if the periosteum was mechanically removed before surgery, as expected (Fig. 5d,

**Fig. 4 | Robust osteogenic capabilities of Fgfr3+ endosteal stromal cells in homeostasis. a** Diagram of lineage-tracing analyses of Fgfr3-creER+ and Gas1-creER+ cells. Right: a proposed model of bone marrow stromal cell differentiation. **b, c** *Fgfr3-creER; R26R^{tdTomato}* (**b**) and *Gas1-creER; R26R^{tdTomato}* (**c**) femur endosteal space at P23 (pulsed at P21) with Col1a1-GFP (left) and Cxcl12-GFP (right). Scale bar: 20 μm. *n* = 4 mice per each group. BM: bone marrow. **d** Flow cytometry analysis of CD45/Ter119/CD31^{neg} bone marrow cells isolated from *Fgfr3-creER; R26R^{tdTomato}* (left) and *Gas1-creER; R26R^{tdTomato}* femurs (center) with *Cxcl12-GFP* (upper) or *Col1a1-GFP* (lower) at P23 (pulsed at P21). *Cxcl12^{GFP/+}; Lepr-cre; R26R^{tdTomato}* femurs at P21 (upper right) Lower right panel: Percentage of tdTomato+ cells among Cxcl12-GFP+ cells (tdTomato expression induced by *Fgfr3-creER, Gas1-creER*, or *Lepr-cre*). Two-tailed, one-way ANOVA followed by Tukey's multiple comparison test. *n* = 5 (Fgfr3^{CE}), *n* = 4 (Gas1^{CE}), *n* = 5 (Lepr^{cre}) mice. **e, f** *Col1a1-GFP; Fgfr3-creER; R26R^{tdTomato}* (**e**) and *Col1a1-GFP; Gas1-creER; R26R^{tdTomato}* (**f**) distal femurs at P49 and 6M (pulsed at P21). Scale bar: 500 μm (left and center right), 20 μm (center left and right). *n* = 4 mice per each time point. BM: bone marrow. **g** Quantification of tdTomato+ bone marrow stromal cells (upper) and osteoblasts and osteocytes (lower) in marrow space at P49 and 6M, aligned based on distance from the growth plate. Fgfr3^{CE}-tdTomato+ cells (red), Gas1^{CE}-tdTomato+ cells (blue). *n* = 4 mice per each group. Two-tailed, Mann–Whitney's *U*-test. **h** Flow cytometry analysis of CD45/Ter119/CD31^{neg} bone marrow cells after 2 days, 1 week, 4 weeks, and 8 weeks of chase (pulsed at P21), isolated from *Fgfr3-creER; R26R^{tdTomato}* (red line) or *Gas1-creER; R26R^{tdTomato}* femurs (blue line). Percentage of lineage-marked tdTomato+ cells among Cxcl12-GFP+ cells (upper) and Col1a1-GFP+ cells (lower). *n* = 3 mice for *Col1a1-GFP/Fgfr3-creER* at P49 and P77, *Col1a1-GFP/Gas1-creER* at P23, *Cxcl12-GFP/ Fgfr3-creER* at P77, *Cxcl12-GFP/Gas1-creER* at P49), *n* = 4 mice for *Col1a1-GFP/Fgfr3-creER* at P23 and P28, *Col1a1-GFP/Gas1-creER* at P28 and P49, *Cxcl12-GFP/Fgfr3-creER* at P28, *Cxcl12-GFP/Gas1-creER* at P23, P28, and P77, *n* = 5 mice for *Col1a1-GFP/Gas1-creER* at P77, *Cxcl12-GFP/Fgfr3-creER* at P23 and P49). Data were presented as mean ± s.d. Exact *P* value is indicated in the figures. Source data are provided as a Source Data file.

second from the left). In contrast, a small number of Fgfr3^{CE}-P21 cells were present within the crater defect if the periosteum was kept intact, and these cells eventually contributed to a small number of osteocytes of the regenerated portion of the cortical bone (Fig. 5d, third and fourth from the left). However, the contribution of Fgfr3+ periosteal cells to cortical bone regeneration was significantly less than that of Fgfr3+ endosteal stromal cells (Fig. 5d, rightmost panel). Second, we utilized a bone marrow ablation model that mechanically removes a defined cylindrical area using an endodontic instrument[40] to induce direct differentiation of marrow stromal cells within the marrow space without involving the periosteum (Fig. 5e). Marrow ablation at P28 induced massive and rapid expansion of Fgfr3^{CE}-P21 cells from the endosteal space after 5 days of injury (Fig. 5e), with the number of Fgfr3^{CE}-P21 cells on the ablated side increased significantly after 7 days of injury (Fig. 5e). Fgfr3^{CE}-P21 cells eventually differentiated into Col1a1(2.3 kb)-GFP+ osteoblasts after 14 days of injury (Fig. 5e). Therefore, Fgfr3+ endosteal stromal cells provide a robust cellular source of osteoblasts during injury responses.

Additionally, we also labeled Fgfr3+ endosteal stromal cells at 9M and performed drill-hole surgery to induce regenerative reactions in these cells. After 7 days of injury, Fgfr3^{CE}-9M stromal cells invaded the injury side as expected (Fig. 5f). However, interestingly, no Fgfr3^{CE}-9M contributed to osteoblasts and osteocytes of the regenerated portion of the cortical bone after 8 weeks of injury (Fig. 5f). Therefore, we postulate that Fgfr3+ endosteal stromal cells lose their osteogenic potential toward adult and aged stages.

## Self-renewal and trilineage potential of Fgfr3+ endosteal stem cells in transplantation

The data shown in Fig. 3c demonstrate that Fgfr3+ endosteal stem cells with long-term ex vivo self-renewability represent a small subset (~1.7%) of Fgfr3-creER+ endosteal stromal cells. We further determined whether Fgfr3+ endosteal stem cells possess OCT potentials and contribute to active osteogenesis in vivo. For this purpose, we transplanted single cell-derived clones of Fgfr3^{CE}-tdTomato+ endosteal stem cells into the femoral marrow cavity of immunodeficient NSG recipient mice (Fig. 6a). After transplantation, Fgfr3^{CE}-tdTomato+ cells engrafted on the endosteal surface, abundantly expressed both ALPL and ACAN and differentiated into osteocytes (Fig. 6b). This indicates that Fgfr3+ endosteal stem cells demonstrate osteoblast-chondrocyte transitional (OCT) identity after homing to their niche following transplantation.

We subsequently performed the cortical bone drill-hole surgery on the recipient mice to induce the regenerative activity of Fgfr3+ endosteal stem cells in the bone marrow (Fig. 6a). Fgfr3^{CE}-tdTomato+ cells contributed to the regenerated portion of the cortical bone and differentiated into ALPL+ osteoblast-like cells within the hard callus

after 2 weeks of injury (Fig. 6c), indicating that Fgfr3+ endosteal stem cells participate in reparative osteogenesis.

We subsequently induced a complete fracture of the transplanted femur to interrogate the in vivo differentiation potential of Fgfr3+ endosteal stem cells. After 14 days of fracture, Fgfr3^{CE}-tdTomato+ cells differentiated into ACAN+ chondrocytes in the cartilaginous fracture callus and ALPL+ osteoblast-like cells in the hard callus (Fig. 6d), indicating that Fgfr3+ endosteal stem cells can become chondrocytes and osteoblasts in vivo. Interestingly, many Fgfr3^{CE}-tdTomato+ cells were localized to the transitional zone between the cartilage and the bone, expressing both ALPL and ACAN, and therefore maintaining osteoblast-chondrocyte transitional identity (Fig. 6d, arrows). These single cell-derived Fgfr3^{CE}-tdTomato+ cells also differentiated into LepR+ reticular cells and Perilipin+ marrow adipocytes (Fig. 6e).

To determine whether Fgfr3+ endosteal stem cells can self-renew in vivo, we further performed secondary CFU-F assays of the transplanted femurs. Fgfr3^{CE}-tdTomato+ cells that engrafted in the endosteal space of the recipient mice formed secondary colonies upon ex vivo isolation and culture at clonal density. Interestingly, all the Fgfr3^{CE}-tdTomato+ secondary clones survived beyond Passage 8, indicating in vivo self-renewal of Fgfr3+ endosteal stem cells (Fig. 6f). Furthermore, these Fgfr3^{CE}-tdTomato+ secondary clones successfully engrafted to the endosteal space of NSG mice and produced osteocytes after 4 weeks of transplantation (Fig. 6g).

Taken together, these findings demonstrate that Fgfr3+ endosteal stem cells can engraft to the endosteal space upon transplantation and self-renew in vivo, while possessing OCT and trilineage potential—particularly under regenerative conditions.

## Fgfr3+ endosteal stromal cells develop aggressive osteosarcoma-like lesions upon p53 loss

Next, we asked if Fgfr3+ endosteal stromal cells could undertake aberrant osteogenic fates in pathological conditions. Osteosarcoma is a bone tumor with an unknown cellular origin, in which p53 (encoded by *Trp53*) is identified as a driver gene in humans and mice[41]. To test the hypothesis that Fgfr3+ endosteal stromal cells can initiate bone tumors, we deleted p53 in Fgfr3+ cells using *Trp53*-floxed alleles. We pulsed cohorts of *Fgfr3-creER; Trp53^{fl/+}; R26R^{tdTomato}* (Fgfr3-p53 Control), *Fgfr3-creER; Trp53^{fl/fl}; R26R^{tdTomato}* (Fgfr3-p53 cKO) at P21 and analyzed these mice at 9 months of age when half of the Fgfr3-p53 cKO mice dropped out from the study due to excessive tumor burden (Fig. 7a).

Strikingly, a substantial fraction of Fgfr3-p53 cKO mice (5/17 mice) developed highly-trabecularized tumors radiating from the marrow space, which destroyed the pre-existing cortical bones (three-dimensional microCT analyses shown in Fig. 7b). Histologically, the tumor cells of Fgfr3-p53 cKO bones were entirely composed of Fgfr3^{CE}-P21Δp53 tdTomato+ cells, characterized by atypical nuclei (Fig. 7c, d and Supplementary Fig. 7a). Moreover, these tumors were composed of

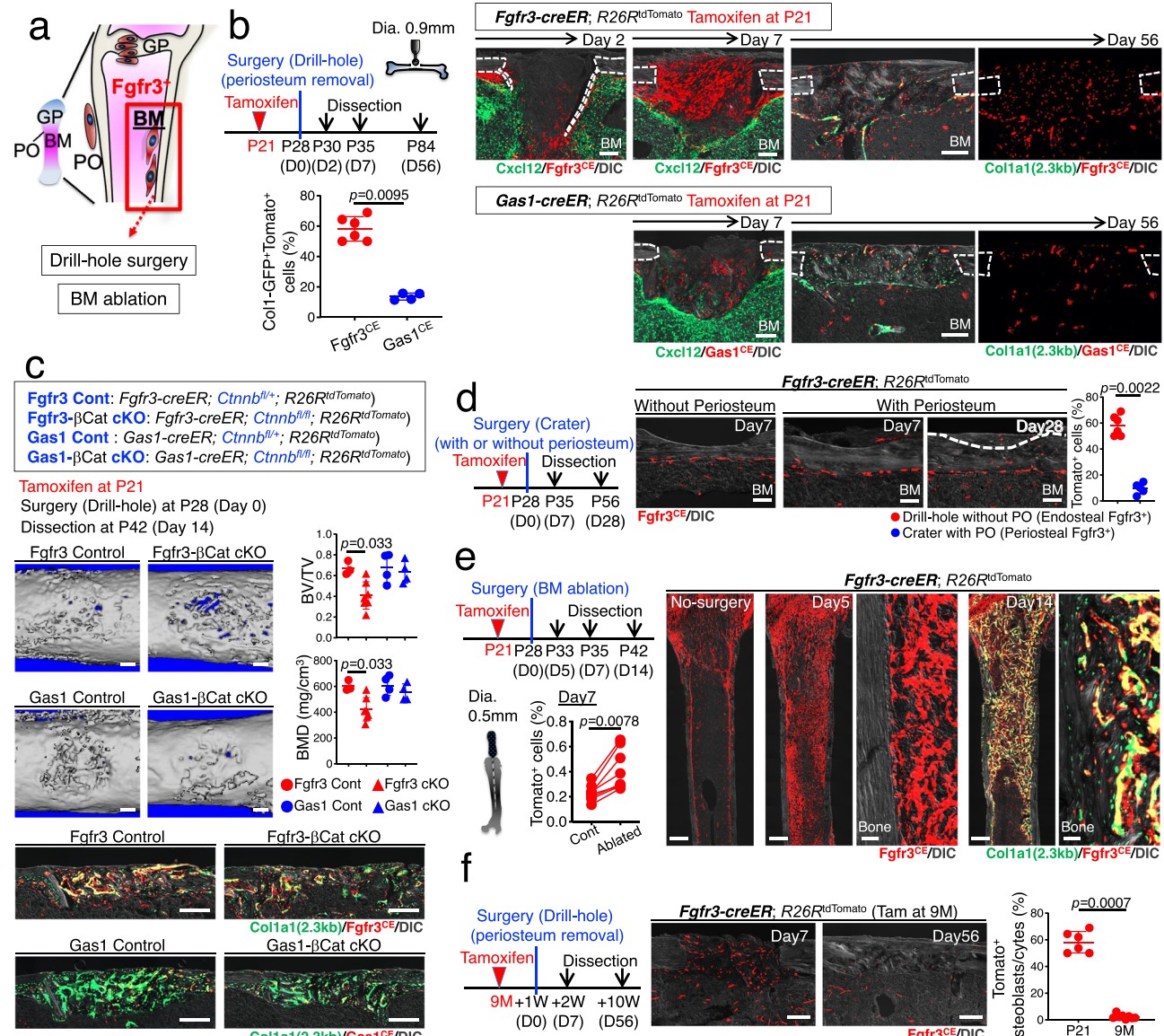

**Fig. 5 | Fgfr3⁺ endosteal stem cells dictate active osteogenesis in injury.**
**a** Diagram of experimental approaches to interrogate Fgfr3⁺ endosteal stromal cells in injury. **b** Drill-hole injury experiments. Timeline and histological analyses. *Fgfr3-creER; R26R^tdTomato* (upper) or *Gas1-creER; R26R^tdTomato* (lower) cortical bone at 2, 7, and 56 days after surgery with *Cxcl12-GFP* (Day 2 and 7) or *Col1a1-GFP* (Day 56). BM: bone marrow. Scale bar: 200 μm. *n* = 6 (Fgfr3^CE at day 56), *n* = 4 (others). Lower left: percentage of Col1a1-GFP⁺tdTomato⁺ cells among total Col1a1-GFP⁺ cells at 56 days after surgery. Two-tailed, Mann−Whitney's *U*-test. **c** Conditional inactivation of Wnt/β-catenin signaling in drill-hole injury. 3D-μCT images (upper) and histological images (lower) of Fgfr3 Control, Fgfr3-βCat cKO, Gas1 Control, and Gas1-βCat cKO injured femur diaphyseal cortical bones at 14 days after surgery. Bone volume/tissue volume (BV/TV) and bone mineral density (BMD) of the injured area (right). Scale bar: 200 μm. *n* = 3 (Fgfr3 Control), *n* = 5 (Fgfr3-βcat cKO), *n* = 4 (Gas1 Control, Gas1-βcat cKO) mice. Two-tailed, Mann−Whitney's *U*-test. **d** Cortical surface crater injury experiments. Injured *Fgfr3-creER; R26R^tdTomato* cortical bone at 7 days (with or without periosteum) and 28 days (with periosteum) after surgery. BM: bone marrow. Scale bar: 200 μm. *n* = 7 (day 7 without periosteum), *n* = 4 (day 7 with

periosteum), *n* = 5 (day 28 with periosteum). Right: Percentage of tdTomato⁺ osteocytes within regenerated bones at day 56 (red: drill-hole from Fig.4b) and 28 (blue: crater with periosteum). Two-tailed, Mann−Whitney's *U*-test. **e** Bone marrow ablation experiments for *Fgfr3-creER; R26R^tdTomato* femurs. Timeline and histological analyses. Histology: Contralateral no-surgery control bone at day 0 (left). Ablated bone at 5 days after surgery (center 2 panels). Ablated bone with *Col1a1-GFP* at 14 days after surgery (right 2 panels). Scale bar: 500 μm (left, left center, right center), 20 μm (center, right). *n* = 4 mice per each group. Percentage of tdTomato⁺ cells in ablated or control femurs at 7 days after surgery (lower left). *n* = 8 mice. Two-tailed, Wilcoxon matched-pairs signed-rank test. **f** Drill-hole injury experiments for 9-month-old *Fgfr3-creER; R26R^tdTomato* bone at 7 and 56 days after surgery. Scale bar: 200 μm. *n* = 4 (day 7), *n* = 8 (day 56). Right: Percentage of tdTomato⁺ osteocytes within regenerated cortical bones at day 56 (young mice from Fig.4b and 9-month-old mice). Two-tailed, Mann−Whitney's *U*-test. Data were presented as mean ± s.d. Exact *P* value is indicated in the figures. Source data are provided as a Source Data file.

woven bones containing abundant TRAP⁺ osteoclast-like cells and rich in Safranin O⁺ chondroid matrices in the central region that chondrocyte-like cells with atypical nuclei occupied (Fig. 7c and Supplementary Fig. 7b). Importantly, the growth plates of Fgfr3-p53 cKO bones were intact (Fig. 7c and Supplementary Fig. 7b), indicating that these bone tumors were not derived from growth plate chondrocytes.

In all the Fgfr3-p53 cKO bones examined (17/17 mice), the entire marrow space was dominated by unorganized trabecular bone-like structures, in which Fgfr3^CE-P21Δp53 tdTomato⁺ cells differentiated into Col1a1(2.3 kb)-GFP⁺ osteoblasts (Fig. 7e). These tumor cells eroded the endocortical surface with typical scalloping patterns (Fig. 7f). Importantly, these Fgfr3^CE-P21Δp53 tdTomato⁺ cells were not

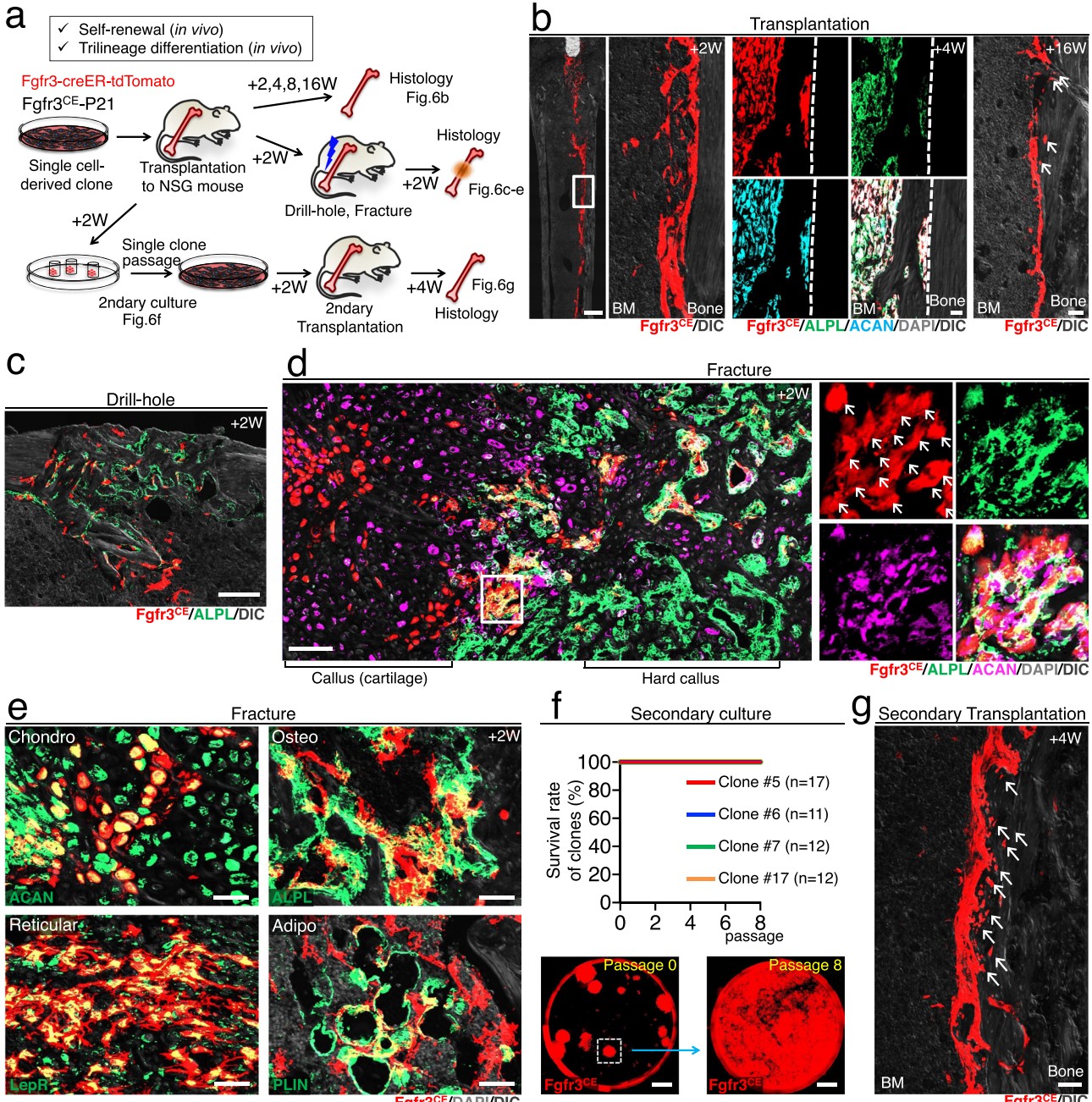

**Fig. 6 | Self-renewal and trilineage potential of Fgfr3⁺ endosteal stem cells in transplantation. a** Diagram of transplantation experiments of ex vivo expanded single cell-derived Fgfr3⁺ endosteal stem cells, involving NSG recipient mice, subsequent surgeries, secondary CFU-F, and transplantation assays. **b** NSG recipient femur transplanted with *Fgfr3-creER*⁺ endosteal stem cells after 2 weeks (left), 4 weeks with ALPL and ACAN staining (center), and 16 weeks (right). Scale bar: 500 μm (left), 20 μm (center), 40 μm (right). *n* = 4 clones. **c** Drill-hole injury of NSG recipient mice transplanted with *Fgfr3-creER*⁺ stem cells, at 2 weeks after surgery with ALPL staining. Scale bar: 200 μm. *n* = 4 clones. **d, e** Complete fracture of NSG recipient mice transplanted with *Fgfr3-creER*⁺ stem cells, at 2 weeks after fracture with ALPL and ACAN staining (**d**, **e**), with LepR and PLIN staining (**e**). Arrows: tdTomato, ALPL, and ACAN triple-positive cells. Scale bar: 100 μm (**d**), 20 μm (**e**). *n* = 4 clones. **f** Secondary CFU-F assays. The survival rate of individual tdTomato⁺ clones harvested from NSG recipient mice with *Fgfr3-creER*⁺ stem cell transplantation (top). Representative images of passages 0 and 8 (bottom). Scale bar: 5 mm. *n* = 4 clones (#5: *n* = 17, #6: *n* = 11, #7: *n* = 7, #17: *n* = 12,). **g** Secondary transplantation of *Fgfr3-creER*⁺ stem cells to NGS recipient femur, at 4 weeks after transplantation. Arrows: osteocytes derived from transplanted tdTomato⁺ stem cells. Scale bar: 20 μm. *n* = 4 clones.

precursor cells of osteoclasts, as these cells did not express Cathepsin K (Fig. 7f).

To further define the importance of Fgfr3⁺ endosteal stromal cells as a potential cell-of-origin of osteosarcoma-like lesions, we performed comparative p53-deficiency-induced bone tumorigenic analyses with additional bone-related *cre/creER* lines, including *Osx-creER* (targeting osteoblast precursors), *Gli1-creER* (targeting growth plate chondrocytes and metaphyseal mesenchymal progenitors), *Pthrp-creER* (targeting resting-zone chondrocytes) and *Lepr-cre* (targeting bone marrow stromal cells). Histologically, only localized pre-tumorous lesions were found in Gli1-p53 cKO (metaphysis) and Osx-p53 cKO (diaphysis) mice, while Pthrp-p53 cKO and Lepr-p53 cKO mice did not develop apparent lesions (Fig. 7g). Importantly, all these p53 cKO mice survived at 9 months (Osx-p53 cKO: 14/14 mice; Gli1-p53

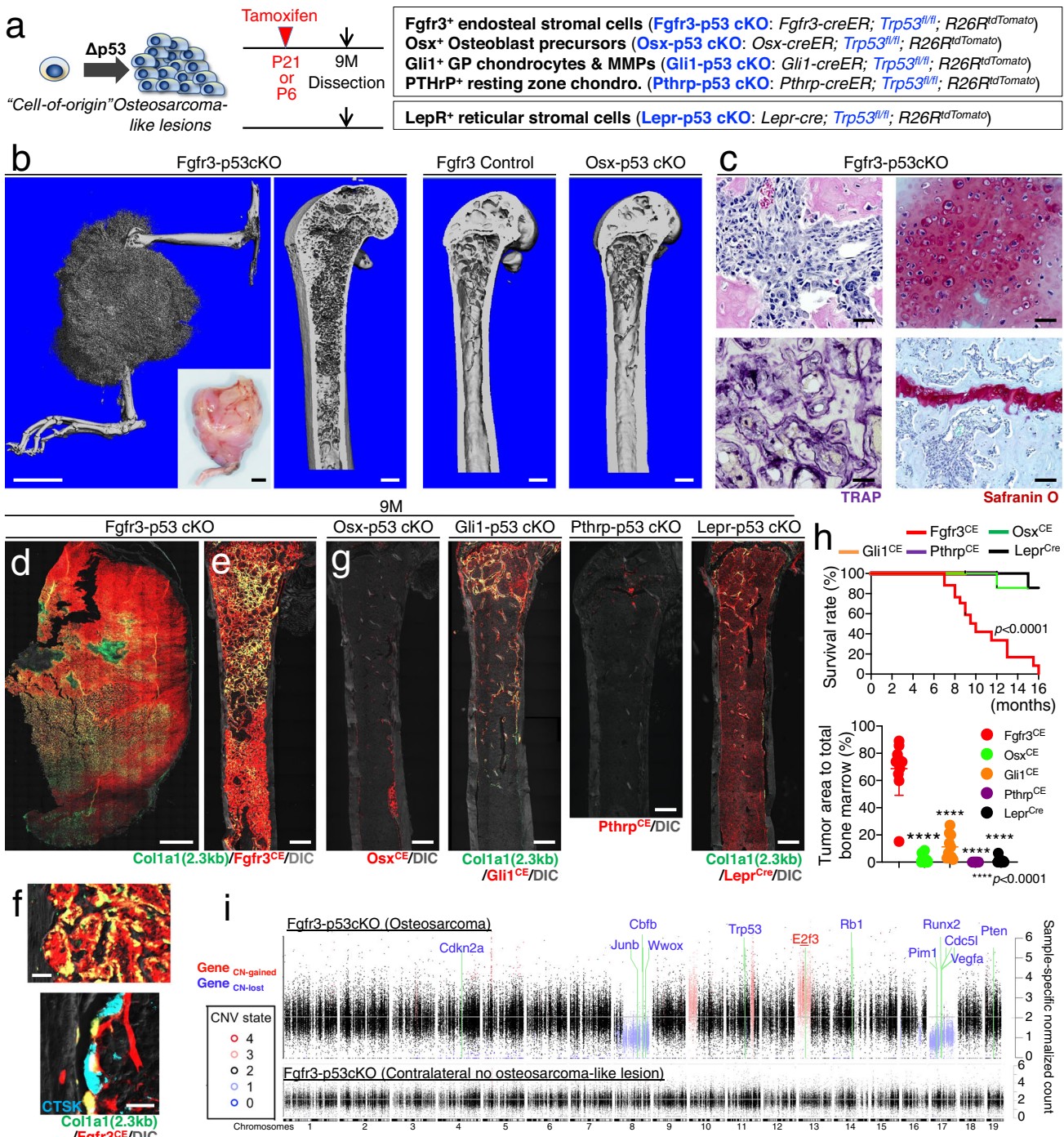

Nature Communications | (2023)14:2383                                                                        12

cKO: 5/5 mice; Pthrp-p53 cKO: 7/7 mice; Lepr-p53 cKO: 13/13 mice), unlike Fgfr3-p53 cKO mice−half of which succumbed at the same stage (Fig. 7h, upper panel). While *Osx-cre; TrpS3^fl/fl* mice are widely accepted as a model for osteosarcoma[42], Osx-p53 cKO bones did not develop radiographically apparent tumorous lesions at this stage (Fig. 7b). We further quantified the area of tdTomato⁺ tumorous lesions relative to the total bone marrow space in the above-described models. Fgfr3^CE-P21Δp53 tdTomato⁺ cells occupied the majority of the bone marrow at 9M (68.7 ± 19.7% of the marrow), while other Δp53 tdTomato⁺ cells occupied much smaller portions of the bone marrow at this stage (Osx^CE: 1.9 ± 2.9%; Gli1^CE: 11.2 ± 9.0%; Pthrp^CE: 0.0 ± 0.0%; Lepr^Cre: 0.98 ± 2.0% of the marrow, Fig. 7h, lower panel).

Whole-exome sequencing (WES) analyses of the Fgfr3-p53 cKO bone tumor revealed widespread copy number variations across the genome, including large-scale deletions on chromosomes 8 and 17, as well as large-scale amplifications on chr10, chr11, and chr13 (Fig. 7i), consistent with the widespread genomic instability typically observed in osteosarcoma. The deletion on chr17 included *Runx2*, a key regulator of bone development, and *Vegfa*. Furthermore, we detected focal deletions of key tumor suppressor genes, including *Rb1* and *Pten*. The analysis was sufficiently sensitive that we even detected the partial deletion of *Trp53*−a floxed region less than 10 kbp−induced by *cre*-mediated recombination in the Fgfr3-p53 cKO lesion. The fact that we detected no significant copy number defects in the contralateral bone with no osteosarcoma-like lesion serves as a strong negative control, confirming the veracity of the copy number defects we found in the tumor bone from the same mouse.

**Fig. 7 | Fgfr3⁺ endosteal stromal cells develop aggressive osteosarcoma-like lesions upon p53 loss. a** Identification of potential cell-of-origin of osteosarcoma-forming cells by comparative p53-deficiency-induced bone tumorigenesis assays. **b–h** Formation of osteosarcoma-like lesions by inducible p53 loss in Fgfr3-creER⁺ cells. **b** Representative 3D-µCT images of Fgfr3-p53 cKO hind limb (left) and femur (center left), *Fgfr3-creER; TrpS3fl/+; R26RtdTomato* (Fgfr3 Control) femur (center right), and Osx-p53 cKO femur (right) at 9M (pulsed at P21). Inset: Gross appearance of the osteosarcoma-like lesion in Fgfr3-p53 cKO hind limb. Scale bar: 5 mm (left, inset), 500 µm (others). *n* = 5 (Fgfr3-p53 cKO, Fgfr3 Control), *n* = 4 (Osx-p53 cKO) mice. **c** Representative H&E (upper left), TRAP (lower left), and Safranin O staining (right) showing tumor cells with large and atypical nuclei and intact growth plates. Scale bar: 50 µm. *n* = 4 mice. **d, e** Representative fluorescent images of Fgfr3-p53 cKO hind limb (**d**) and femur (**e**) with *Col1a1-GFP*. Scale bar: 2 mm (**d**), 500 µm (**e**). *n* = 11 mice. **f** Fgfr3-p53 cKO bone marrow with endocortical scalloping (upper) and Cathepsin K immunostaining (lower). Scale bar: 20 µm. *n* = 4 mice. **g** Lineage-tracing analyses of p53-deficient osteosarcoma-forming cells. Osx-p53 cKO, Gli1-p53 cKO (pulsed at P21), Pthrp-p53 cKO (pulsed at P6), and Lepr-p53 cKO femur at 9M. Scale bar:

500 µm. *n* = 13 (Osx-p53 cKO), *n* = 10 (Gli1-p53 cKO), *n* = 8 (Pthrp-p53 cKO), *n* = 12 (Lepr-p53 cKO) femurs. **h** Survival rate of various p53 cKO mice (upper). *n* = 17 (Fgfr3-p53 cKO), *n* = 14 (Osx-p53 cKO), *n* = 5 (Gli1-p53 cKO), *n* = 7 (Pthrp-p53 cKO), *n* = 13 mice (Lepr-p53 cKO). Log-rank (Mantel–Cox) test. Quantification of tumor area relative to total marrow space in distinct cKO at 9M (lower). *n* = 11 (Fgfr3-p53 cKO), *n* = 13 (Osx-p53 cKO), *n* = 10 (Gli1-p53 cKO), *n* = 8 (Pthrp-p53 cKO), *n* = 12 (Lepr-p53 cKO) femurs. Two-tailed, one-way ANOVA followed by Tukey's multiple comparison test. **i** Pairwise mWES CNV analyses. Fgfr3-p53 cKO osteosarcoma-like lesions (upper). Fgfr3-p53 cKO contralateral bone without osteosarcoma-like lesion (lower). Background control: *TrpS3fl/fl* bone. X-axis: Autosomal chromosome position. Y-axis: Normalized count. Black dots: regions with a normal copy-number state. Blue dots: copy-number lost regions. Red dots: copy-number gained regions. Positions of known oncogenes and tumor suppressor genes with copy number changes are indicated with green lines. Gene symbols are colored by copy number status. Data were presented as mean ± s.d. Exact *P* value is indicated in the figures. Source data are provided as a Source Data file.

Therefore, these findings identify Fgfr3⁺ endosteal stromal cells as a potent cellular source of aggressive osteosarcoma-like lesions, supporting the notion that osteosarcoma may preferentially arise from a stem/stromal cell population in the endosteal space.

## Single-cell characterization of Fgfr3⁺ cell-derived mouse osteosarcoma-like lesions

We further defined the molecular characteristics of these osteosarcoma-like lesions at the single-cell level. To this end, we investigated how p53 loss induces a change in cell populations and their transcriptomes by combined lineage-tracing and single-cell RNA-seq analyses. We interrogated the single-cell transcriptomic profiles of lineage-marked Fgfr3CE-P21 Control tdTomato⁺ cells (isolated from *Fgfr3-creER; TrpS3fl/+; R26RtdTomato* bones, pulsed at P21) and Fgfr3CE-P21Δp53 tdTomato⁺ cells (isolated from *Fgfr3-creER; TrpS3fl/fl; R26RtdTomato* bones, pulsed at P21) at 4M, then integrated the two datasets using LIGER (Fig. 8a).

Initial analysis revealed a distinct group of chondrocyte-like cells (Chondro) that were separate from other major cell clusters constituted by osteoblastic cells (Osteo), pre-adipocyte-like reticular cells (Adipo), and osteoblast-reticular transitional (ORT) cell types (Fig. 8b). The group of chondrocyte-like cells significantly expanded in the Δp53 sample and were predominantly composed of Δp53 cells (Fig. 8c, d). In addition, a small number of Δp53 OCT stem cells demonstrated an aberrant upregulation of a chondrocyte marker *Acan* and a positive cell cycle regulator *Mki67*, as well as a downregulation of an osteoblast marker *Col1a1* (Fig. 8e), suggesting that Δp53-induced gene expression changes may occur in rare stem-like cells.

We further interrogated the tumor cell characteristics of these aberrant chondrocyte-like cells with CopyKAT[43] and InferCNV (https://github.com/broadinstitute/inferCNV), computational packages that infer copy number variation based on single-cell expression data. These analyses revealed that the chondrocyte-like cells in the Δp53 sample were predominantly aneuploid (Fig. 8f). These aneuploid cells showed significant copy number changes across the entire genome, including both gain and loss (Fig. 8g). In addition, the aneuploid chondrocyte-like tumor cells contained subpopulations with different patterns of copy number variation (Fig. 8h), suggesting intratumoral cellular heterogeneity. To further investigate the characteristics of the aberrant chondrocyte-like cells, we identified genes differentially expressed between the Δp53 and Control cells within this specific population. Gene Ontology term enrichment analysis revealed several gene functions consistent with tumor cells, including cell cycle regulation and cellular response to external signals (Fig. 8i and Supplementary Fig. 8a). These results also suggest aberrant regulation of bone cell differentiation, with upregulation of genes related to positive regulation

of cell differentiation in Δp53, and upregulation of genes related to negative regulation of cell differentiation in Control.

To further confirm the similarity between the chondrocyte-like cells in our mouse model and osteosarcoma cells, we integrated our dataset with a previously published single-cell RNA-seq dataset of human chondroblastic osteosarcoma[44] (Fig. 8j and Supplementary Fig. 8b). This cross-species comparative analysis confirmed that the chondrocyte-like putative tumor cells in our mouse Δp53 tumor model shared strong transcriptional similarity with those from human osteosarcoma samples (Fig. 8j and Supplementary Fig. 8c). The chondrocyte-like and osteoblast-like cells from the mouse data aligned with the chondrocyte-like and osteoblast-like cells from the human sample, and were present in similar proportions in both species. In contrast, the reticular-like cells from the mouse sample did not align with any cells in the human sample, indicating that the computational analysis is not spuriously aligning non-corresponding cells. Overall, these results suggest that our mouse model recapitulates key features of human osteosarcoma.

Together, these integrative single-cell computational analyses provide evidence that p53 loss in Fgfr3⁺ endosteal stromal cells leads to osteosarcoma-like lesions associated with genomic instability and intratumoral heterogeneity, which share strong transcriptional similarity with chondroblastic osteosarcoma in humans.

## Unregulated self-renewal and aberrant osteogenic fates of tumorigenic Fgfr3⁺ stem cells

Lastly, we set out to define the cellular mechanisms underlying the tumorigenicity of Fgfr3⁺ endosteal stem cells. For this purpose, we performed CFU-F assays of Fgfr3CE-P21 Δp53 and Fgfr3CE-4M Δp53 cells by plating bone marrow cells isolated from *Fgfr3-creER; TrpS3fl/fl; R26RtdTomato* mice at P28 and 4M (pulsed at P21) at clonal density. Strikingly, Fgfr3CE-Δp53 tdTomato⁺ clones exhibited substantially faster cell growth and higher passageability (Fig. 9a, b); as many as 82.9% of Fgfr3CE-P21Δp53 clones could reach Passage 8 with a duration of 74.0 ± 0.0 days; in contrast, Fgfr3CE-P21 wild-type tdTomato⁺ clones took a significantly longer duration of 386.4 ± 91.4 days to reach Passage 8 (Fig. 9c). This data indicates that Fgfr3⁺ CFU-Fs uniformly acquire skeletal stem cell-like properties upon p53 loss associated with higher clonogenicity.

We further examined the transcriptomes of Fgfr3CE-Δp53 clones using RNA-seq. Principal component analysis revealed that the transcriptomes of Fgfr3CE-Δp53 clones at P21 and 4M exhibited a distinct pattern from those of Fgfr3CE-WT clones on PC1 (Fig. 9d), demonstrating that p53 loss induced consistent changes of gene expression in Fgfr3⁺ endosteal stem cells (see Supplementary Fig. 9 for evidence of p53 loss). Interestingly, Fgfr3CE-WT clones were more variable than Fgfr3CE-Δp53 clones on PC2 (Fig. 9d), indicating that

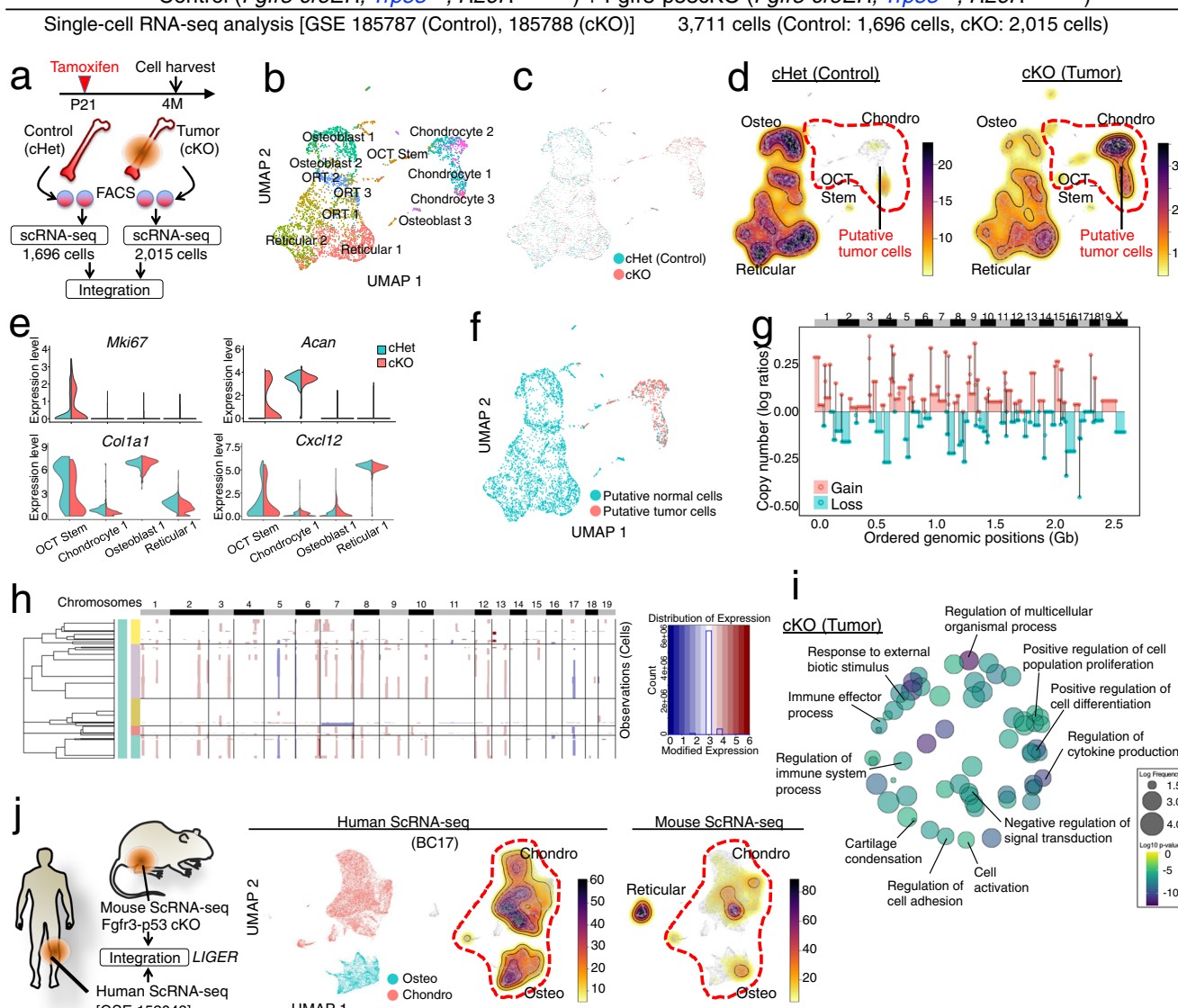

**Fig. 8 | Single-cell characterization of Fgfr3⁺ cell-derived mouse osteosarcoma-like lesions. a–e** Single-cell RNA-seq analyses of osteosarcoma-like lesions. **a** Diagram for LIGER data integration of single-cell RNA-seq datasets of lineage-traced tdTomato⁺ cells isolated from *Fgfr3-creER; Trp53^fl/+^; R26R^tdTomato^* (Fgfr3-p53 cHet) and *Fgfr3-creER; Trp53^fl/fl^; R26R^tdTomato^* (Fgfr3-p53 cKO) femurs at 4M (pulsed at P21). Fgfr3-p53 cHet: 1696 cells, Fgfr3-p53 cKO: 2015 cells, Total: 3711 cells. **b** UMAP visualization of major sub-clusters of the LIGER-integrated dataset. **c** UMAP visualization of two datasets integrated by LIGER. Blue: cHet, red: cKO. **d** Density plots of Fgfr3-p53 cHet (left) and Fgfr3-p53 cKO (right) in the integrated space. Red dotted contour: OCT Stem and Chondrocyte-like clusters predominantly contributed by Fgfr3-p53 cKO cells. **e** Split violin plots for *Mki67* (proliferation), *Acan* (Chondro), *Col1a1* (Osteo), and *Cxcl12* (Reticular) in representative clusters (OCT Stem, Chondrocyte 1, Osteoblast 1, and Reticular 1) between cHet (blue) and cKO (red). *n* = 148 (OCT Stem), *n* = 26 (Chondro 1), *n* = 290 (Osteo 1), *n* = 469 (Reticular 1) cells in cHet. *n* = 426 (OCT Stem), *n* = 125 (Chondro 1), *n* = 177 (Osteo 1), *n* = 434 (Reticular 1) cells in cKO. **f** UMAP-based visualization of putative aneuploid cells inferred by

CopyKAT. Blue: putative normal cells. Red: putative tumor cells. **g** Genome-wide visualization of copy number variations in putative tumor cells inferred by Copy-KAT. **h** Hierarchical clustering of putative tumor cells with different patterns of copy number variation inferred by inferCNV. **i** REVIGO plot of gene ontology terms enriched among genes upregulated in putative tumor cells. The *p* values were calculated using GOrilla's one-sided test based on the hypergeometric distribution, with a flexible *p* value cutoff applied for multiple testing correction. The results were subsequently transferred to REVIGO for visualization. **j** Mouse-human tumor data integration. Left: Diagram for LIGER data integration of single-cell RNA-seq datasets from mouse and human osteosarcoma cells. Center to the right: Fgfr3-p53 cKO scRNA-seq dataset was integrated with the human chondroblastic osteosarcoma scRNA-seq dataset (BC17). UMAP visualization of human chondroblastic and osteoblastic osteosarcoma cells in the integrated space (center left). Density plots of human (BC17) (center right) and mouse (right) tumor cells in the integrated UMAP space.

that Fgfr3⁺ SSC clones may undergo phenotypic convergence due to p53 loss and carry similar molecular characteristics. Specifically, Fgfr3^CE^-Δp53 clones showed increased expression levels of osteoblast markers, cell proliferation, and tumor-related genes (Fig. 9e); the overall expression level of these genes was higher in Fgfr3^CE^-Δp53 clones at 4M than at P21 (Fig. 9e). Note that these results are also consistent with the single-cell analysis of expression

differences in chondrocyte-like cells between Δp53 and Control cells shown (Fig. 8i and Supplementary Fig. 8a). Therefore, these analyses demonstrate that p53 loss introduces an osteogenic bias to Fgfr3⁺ endosteal stem cells.

We further tested the in vivo transplantability of Fgfr3^CE^-P21Δp53 single cell-derived clones. Fgfr3^CE^-P21Δp53 or Fgfr3^CE^-P21 wild-type tdTomato⁺ clonal cells (10⁵ cells) were transplanted into the

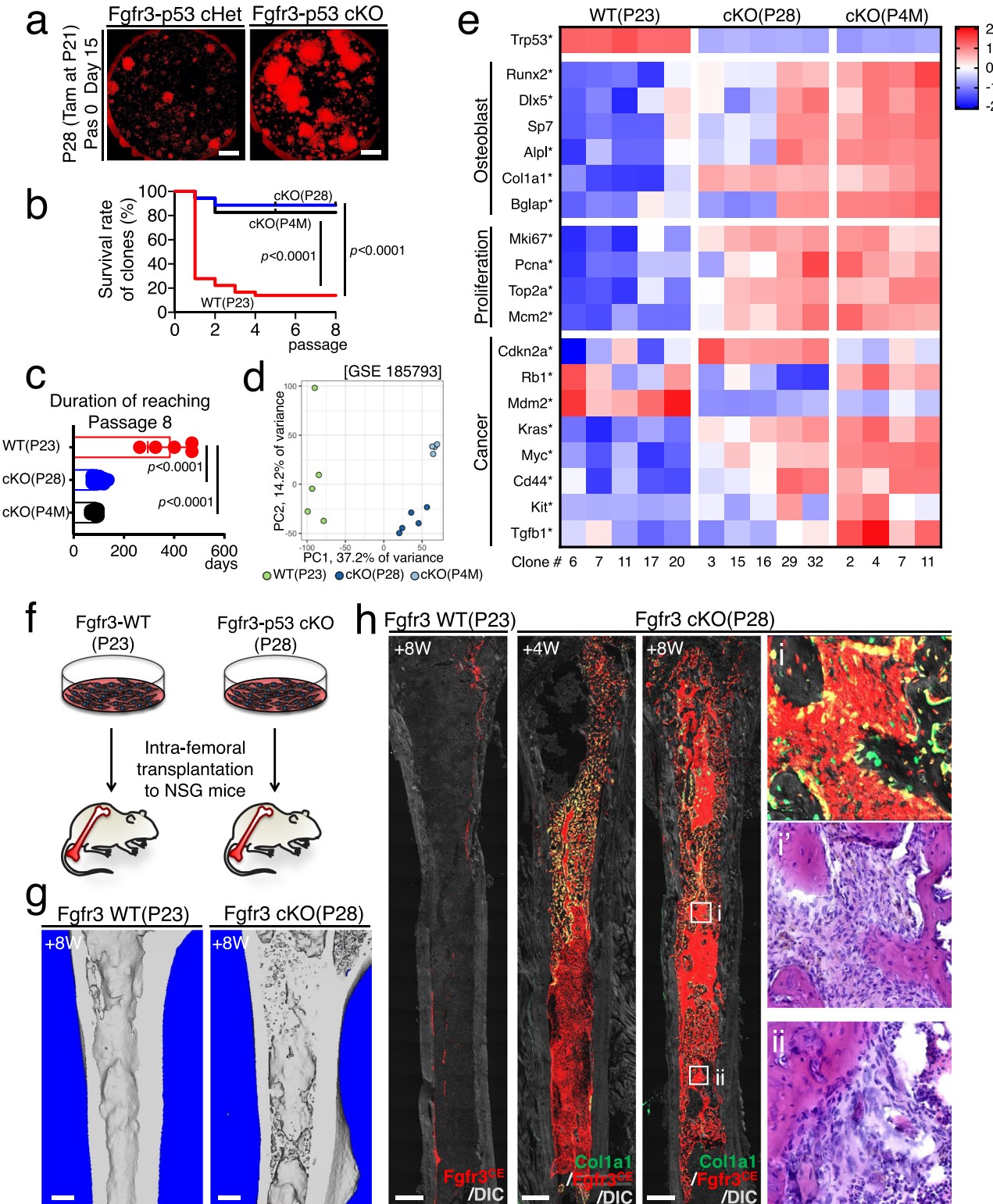

femoral marrow cavity of immunodeficient NSG recipient mice (Fig. 9f). After 4 weeks of transplantation, Fgfr3CE-P21 WT clones engrafted on the endosteal surface; strikingly, Fgfr3CE-P21Δp53 clones massively expanded in the marrow space. After 8 weeks of transplantation, Fgfr3CE-P21Δp53 clones robustly differentiated into Col1a1-GFP+ osteoblasts on the bone surface and generated highly-trabecularized structures throughout the marrow space reminiscent of early-stage osteosarcoma-like lesions, associated with tumor cells with atypical nuclei and endocortical resorption with scalloping patterns (Fig. 9f–h).

Taken together, these findings indicate that Fgfr3+ endosteal stem cells develop aggressive osteosarcoma-like lesion upon p53 tumor suppressors through unregulated self-renewal and aberrant osteogenic fates.

## Discussion

Here, we identified a novel population of bone marrow stromal cells with osteoblast-chondrocyte transitional (OCT) identities in the endosteum, which are abundant in young stages and coordinate active

**Fig. 9 | Unregulated self-renewal and aberrant osteogenic fates of tumorigenic Fgfr3+ endosteal stem cells. a** CFU-F assay of Fgfr3-p53 cHet and Fgfr3-p53 cKO bone marrow cells at P28 (pulsed at P21). Scale bar: 5 mm. *n* = 4 mice per each group. **b** Survival curve of individual tdTomato+ clones over serial passages among Fgfr3-WT (P23, red), Fgfr3-p53 cKO (P28, blue), and Fgfr3-p53 cKO (P4M, black). *n* = 36 (Fgfr3-WT P23), *n* = 35 (Fgfr3-p53 cKO P28), *n* = 18 (Fgfr3-p53 cKO P4M) clones. Log-rank (Mantel–Cox) test. **c** Duration to reach Passage 8. *n* = 5 (Fgfr3-WT P23), *n* = 29 (Fgfr3-p53 cKO P28), *n* = 12 (Fgfr3-p53 cKO P4M) clones. Two-tailed, one-way ANOVA followed by Tukey's multiple comparison test. **d**, **e** Comparative bulk RNA-seq analysis of individual tdTomato+ clones, Fgfr3-WT (P23, *n* = 5 clones), Fgfr3-p53 cKO (P28, *n* = 5 clones) and Fgfr3-p53 cKO (P4M, *n* = 4 clones) clones. **d** Principal component analysis (PCA) plot of Fgfr3-WT (P23, light green), Fgfr3-p53 cKO (P28, blue) and Fgfr3-p53 cKO (P4M, light blue) clones. *x*-axis: PC1, 37.2% variance, *y*-axis: PC2, 14.2% variance. **e** Heatmap of representative differentially expressed genes (DEGs) associated with osteoblast differentiation, cell proliferation, and osteosarcomagenesis. Star: statistically significant difference among the three groups. *n* = 4 (Fgfr3-p53 cKO P4M), *n* = 5 (Fgfr3-WT P23, Fgfr3-p53 cKO P28) mice. A two-sided *t*-test, based on a beta-negative binomial distribution, was used to calculate the *p* values (corrected for multiple testing) with an FDR cutoff of 0.05 to identify DE for the bulk RNA-seq data. **f** Diagram of intrafemoral transplantation of individual Fgfr3-WT (P23) and Fgfr3-p53 cKO (P28) tdTomato+ clones to NSG recipient mice. **g** 3D-μCT of Fgfr3-WT (P23) and Fgfr3-p53 cKO (P28) transplanted femurs after 8 weeks of transplantation. Scale bar: 500 μm. *n* = 4 per each group. **h** Histology of recipient femurs. (Left): Transplantation of Fgfr3-WT (P23) clones after 8 weeks of transplantation. (Left center): Transplantation of Fgfr3-p53 cKO (P28) clones after 4 weeks (left center) and 8 weeks (right center) of transplantation. (Right): Magnified views of highly-trabecularized areas in recipient bone marrow, fluorescent, and H&E staining (right). Scale bar: 500 μm. *n* = 4 per each group. Data were presented as mean ± s.d. Exact *P* value is indicated in the figures. Source data are provided as a Source Data file.

bone formation; these cells can also be converted into osteosarcoma-forming cells upon loss of the p53 tumor suppressor. These "OCT" stem cells in the endosteal space express *Fgfr3*, and are distinct from skeletal stem cells in adult bone marrow defined by *LepR* expression[20] or growth plate chondrocytes that undergo a chondrocyte-to-osteoblast transformation in early life[12,22,26]. These bone marrow endosteal stem/stromal cells have several unique characteristics; first, the cells are positioned in a unique state between osteoblasts and chondrocytes while being primed to become the three types of cells; second, these cells reside in a specialized microenvironment of the bone marrow endosteum; and third, these cells have the robust self-renewal capacity and osteogenic potential, which may render these cells highly susceptible for tumorigenic transformation. We emphasize that our definition of OCT identities does not infer cell type plasticity between two differentiated cell types, wherein cells are transitioning between osteoblasts and chondrocytes. We define OCT identities as a state with some characteristics of both osteoblasts and chondrocytes. Our findings that these OCT cells are particularly abundant in the young bone marrow and depleted in aged bone marrow denote their transitional nature. Together, our findings reveal important new insights into a new class of skeletal stem cells, identifying *Fgfr3*-expressing bone marrow endosteal stromal cells as an important regulator of bone homeostasis.

Maintaining a state epigenomically primed to multiple terminally differentiated types may confer clonal advantages for somatic stem cells. Fgfr3+ endosteal stromal cells that we identified here are characterized by a trilineage potential primed for the three different types of terminally differentiated cells—osteoblasts, chondrocytes, and pre-adipocyte-like reticular cells. This characteristic makes these cells highly responsive to external cues, which can rapidly and robustly produce bone-forming osteoblasts in homeostasis and regeneration. Interestingly, the epigenomic regulatory landscape of these cells is biased toward the reticular lineage, requiring some external signals to become osteoblasts.

The chondrocyte-to-osteoblast transformation has been identified as one of the mechanisms to produce osteoblasts in the endochondral pathway[22,26]. Yet, it remains to be largely determined how efficiently chondrocyte-to-osteoblast transformation occurs, and how significantly this process contributes to osteoblastogenesis, particularly in the postnatal stage when the proliferative activities of the growth plate cartilage markedly decline. Identification of OCT stromal cells may at least partly fill the gap in knowledge, in that partially "chondrocyte-like" cells in young bone marrow can actively participate in osteogenesis. Interestingly, recent large-scale single-cell RNA-seq studies have identified groups of cells annotated as "chondrocytes" among adult bone marrow stromal cells that are, theoretically, free of growth plate chondrocytes[16,19]. Chondrocyte-like osteoprogenitor (COP) cells have been identified among metaphyseal mesenchymal progenitors (MMPs) marked by *Gli1-creER*[37]. These cells may at least in part overlap with Fgfr3+ endosteal stromal cells with osteoblast-chondrocyte transitional identities that we report here. It is possible that undifferentiated skeletal lineage cells can fluctuate among the three terminal states of osteoblasts, chondrocytes, and adipocytes to adeptly respond to various environmental cues and regulate the rate of bone formation.

The uniquely transitional nature may render Fgfr3+ endosteal stromal cells highly susceptible to tumorigenic transformation. Our findings demonstrate that p53 loss confers these stromal cells with unregulated self-renewal capacity and increased osteogenic bias through chondrocyte-like intermediates, endowing the neoplastic cells with aggressive behaviors associated with genomic instability and intratumoral heterogeneity. The cellular origin of osteosarcoma is not well defined[41]. Our findings provide preliminary evidence that osteosarcoma may preferentially arise from skeletal stem cells in young bone marrow, which may partly explain their dominant occurrence in childhood. We acknowledge, however, that increased efficiency of tumorigenesis in a mouse genetic model does not necessarily indicate that we have identified the sole cell-of-origin for aggressive bone tumors in humans.

In conclusion, our findings identify a novel type of skeletal stem cells residing in the bone marrow endosteum that are uniquely positioned to coordinate rapid osteogenesis occurring in young bones. The ambiguous state with epigenomic priming to differentiated cell types may be one of the unique features of skeletal stem cells that are responsible for producing a wide variety of cells of the skeletal cell lineage (Fig. 10).

## Methods

### Mice

*Cxcl12*GFP/+[35] and *Osx-creER*[45] mice have been described previously. *Prrx1-cre* (JAX005584), *Lepr-cre* (JAX008320), *Col2a1-creER* (JAX006774), *Fgfr3-iCreER* (JAX025809), *Gli1-creER* (JAX007913), *Rosa26-CAG-loxP-stop-loxP-tdTomato* (Ai14: *R26R-tdTomato*, JAX007914), *Rosa26-CAG-loxP-stop-loxP-ZsGreen* (Ai6: *R26R*-ZsGreen, JAX007906), *Col1a1*(2.3 kb)-GFP (JAX013134), *Osteocalcin-GFP* (JAX017469), *Osterix-mCherry* (JAX024850), *Ctnnb1-floxed* (JAX004152), *Trp53-floxed* (JAX008462), NOD scid gamma (NSG) (JAX005557), FVB/NJ (JAX001800), and C57BL/6J (JAX000664) mice were acquired from the Jackson Laboratory. *Fgfr3-GFP* (MMRRC:031901-UCD) mice were acquired from the Mutant Mouse Resource and Research Centers. All procedures were conducted in compliance with the Guidelines for the Care and Use of Laboratory Animals approved by the University of Texas Health Science Center at Houston's Animal Welfare Committee (AWC), protocol AWC-21-0070, and the University of Michigan's Institutional Animal Care and Use Committee (IACUC), protocol 9496. All mice were housed in a specific pathogen-free condition, and analyzed in a mixed background. Mice were housed in static microisolator cages (Allentown Caging, Allentown, NJ). Access to water and food (irradiated LabDiet 5008,

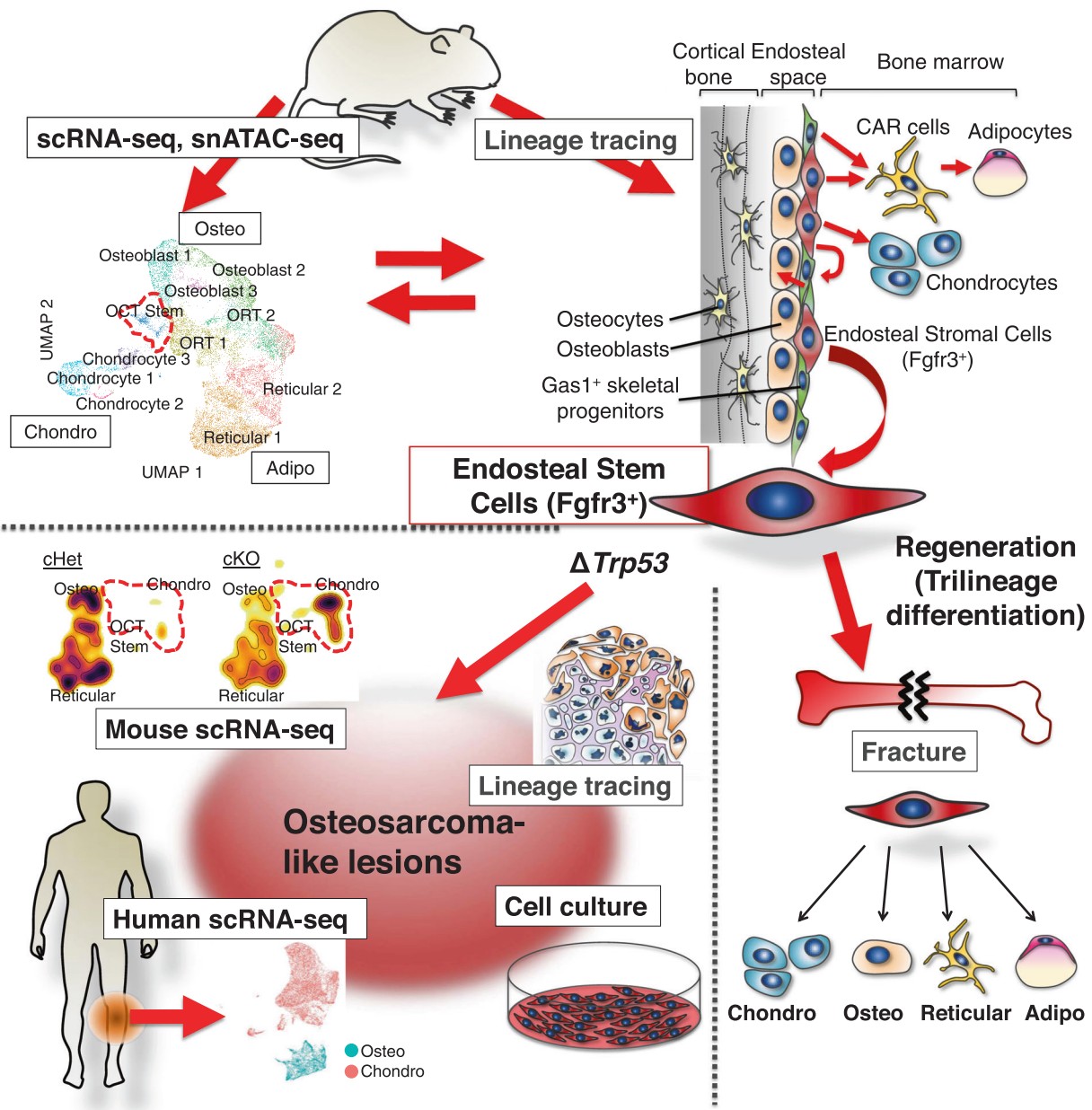

**Fig. 10 | Location and function of Fgfr3-expressing endosteal stem/stromal cells.** Fgfr3-expressing endosteal stem cells with osteoblast-chondrocyte transitional (OCT) identities in young bone marrow dictate active osteogenesis and aggressive tumorigenesis. The ambiguous state with epigenomic priming to differentiated cell types may be one of the unique features of skeletal stem cells that are responsible for producing a wide variety of cells of the skeletal cell lineage in physiological and pathological conditions.

Richmond, IN) was ad libitum. Animal rooms were climate controlled to provide temperatures of 22–23 °C and 40–65% of humidity on a 12 h light/dark cycle (lights on at 0600). Both male and female mice were used for the study, as we did not observe any sex bias. Mice were monitored to ensure tumor burdens did not exceed 10% body weight, 1000 cm³, or 1.5 cm tumor size, as per the AWC guideline. The maximal tumor size/burden was not exceeded in our study. For all breeding experiments, *creER* transgenes were maintained in male breeders to avoid spontaneous germline recombination. Mice were identified by micro-tattooing or ear tags. Tail biopsies of mice were lysed by a Hot-Shot protocol (incubating the tail sample at 95 °C for 30 min in an alkaline lysis reagent followed by neutralization) and used for PCR-based genotyping (GoTaq Green Master Mix, Promega, and Nexus X2, Eppendorf). Perinatal mice were also genotyped fluorescently (BLS miner's lamp) whenever possible. Mice were euthanized by over-dosage of carbon dioxide or decapitation under inhalation anesthesia in a drop jar (Fluriso, Isoflurane USP, VetOne).

### Generation of *Gas1-creER* knockin mice

CRISPR/Cas9 technology was used to generate a knockin mouse that inserts the iCreER^T2 coding sequence in the 3′ UTR region of the *Gas1* gene. CRISPOR.tefor.net (Haeussler et al., 2016) was used to select single guide RNAs (sgRNA) that were predicted to cut near the *Gas1* termination codon. Phosphorothioate-modified sgRNA predicted to be specific and active were synthesized by Synthego.com[46]. sgRNA were used to form ribonucleoprotein (RNP) complexes with enhanced-specificity Cas9 protein (Sigma-Aldrich ESPCAS9PRO)[47]. RNPs were tested to determine if they would cause chromosome breaks upon pronuclear microinjection in mouse zygotes. Briefly, RNP were formed by combining 30 ng/µl of sgRNA with 30 ng/µl of enhanced-specificity

Cas9 protein. RNP was microinjected into mouse zygotes that were then cultured in vitro to blastocyst embryos (~64 cells per embryo). DNA extracted from individual blastocysts was used for PCR. PCR primers amplified a 704 bp genomic DNA fragment, including the sgRNA target (Forward primer: 5′-CATCTGCGAATCGGTCAAAGAGAA-CAT-3′; Reverse primer: 5′-CTTTGCAAACCTAATTCCCTCAACATCC-3′). Amplicons were submitted for Sanger sequencing. The number of blastocyst chromatograms that returned superimposed peaks, typical of indels formed by nonhomologous endjoining repair of Cas9-induced chromosome breaks[48], was counted and used to score chromosome break activity. sgRNA C126G1 (5′-CCTGGAGCAGCCTGCAG-CAG-3′ (PAM = GGG)) was selected to produce knockin mice because 67% of blastocysts showed the presence of indels. A plasmid containing 2 kb arms of homology upstream and downstream of the Cas9 cut site 8 bp downstream of the *Gas1* termination codon was synthesized and the P2A-iCreER^T2-bGH polyA signal sequence was subcloned between the arms (Genscript). The use of the P2A sequence will result in the expression of the GAS1 protein and the iCreER^T2 protein from a single bicistronic mRNA under the control of the *Gas1* regulatory elements[49]. Mouse zygotes were obtained by mating B6SJLF1/J mice (JAX 100012). Zygotes were microinjected with RNP and 10 ng/μl circular DNA donor plasmid and transferred to pseudopregnant B6D2F1/J mice (JAX100006). After 64 possible G0 founder pups were born, DNA was extracted from tail tip biopsies and amplified with primers to detect correct insertion. Five G0 founders were identified to mate with wild-type mice for germline transmission of the *Gas1-P2A-iCreER^T2-bGH polyA* knockin allele. The use of KOD hot start polymerase and 5XCES in PCR (Ralser et al. 2006) were critical for success, along with increasing the GC content of the primers to 55%[50].

### Tamoxifen

Tamoxifen (Sigma T5648) was mixed with 100% ethanol until completely dissolved. Subsequently, a proper volume of sunflower seed oil (Sigma S5007) was added to the tamoxifen-ethanol mixture and rigorously mixed. The tamoxifen-ethanol-oil mixture was incubated at 60 °C in a chemical hood until the ethanol evaporated completely. The tamoxifen-oil mixture was stored at room temperature until use. Tamoxifen was injected into mice intraperitoneally using a 26-1/2-gauge needle (BD309597). Tamoxifen was injected at a dose of 1 mg at P21 or 2 mg at 8 weeks and 9 months of age.

### Surgery

Mice were anesthetized through a nose cone, in which 1.5–2% isoflurane was constantly provided with oxygen through a vaporizer. For drill-hole injury, left femurs were operated, while right femurs were untreated and used as an internal control. An incision was made in the lateral skin of the mid-femur. After splitting the muscle, the periosteum was separated to expose the femoral surface. A drill-hole injury was made onto the diaphysis by a Micro-Drill (Stoelting) with a 0.9-mm diameter stainless steel burr (Fine Science Tools). The periosteum of the surgical field was scraped off using a scalpel prior to the procedure.

For femoral complete fracture surgery, a small incision was made in the lateral skin of the mid-femur. After splitting the muscle, a non-stabilized fracture was created in the diaphysis using surgical scissors (Miltex). The surgical field was irrigated with saline, and the incision line was sutured.

### Histology and immunohistochemistry

Samples were fixed in 4% paraformaldehyde for a proper period, typically overnight at 4 °C, then decalcified in 15% EDTA for a proper period, typically ranging from 1 to 14 days. Decalcified samples were cryoprotected in 30% sucrose/PBS solutions and then in 30% sucrose/PBS:OCT (1:1) solutions, each at least overnight at 4 °C. Samples were embedded in an OCT compound (Tissue-Tek, Sakura) and cryosectioned at 14 μm using a cryostat (Leica CM1850) and adhered to positively charged glass slides (Fisherbrand ColorFrost Plus). Sections were postfixed in 4% paraformaldehyde for 15 min at room temperature. Then, stained with hematoxylin and eosin, Safranin O and Fast Green, and Tartrate-resistant acid phosphatase (TRAP).

For immunostaining, sections were permeabilized with 0.25% TritonX/TBS for 30 min, blocked with 3% BSA/TBST for 30 min, and incubated with goat anti-ALPL polyclonal antibody (1:100, R&D, AF2910), goat anti-GAS1 polyclonal antibody (1:200, R&D, AF2644), goat anti-Gremlin polyclonal antibody (1:200, R&D, AF956), goat anti-LepR polyclonal antibody (1:200, R&D, AF497), rabbit anti-FGFR3 polyclonal antibody (1:200, Santa Cruz, sc-123), rabbit anti-Aggrecan polyclonal antibody (1:200, Sigma, AB1031), rabbit anti-perilipin A/B polyclonal antibody (1:500, Sigma, P1873), or rabbit anti-Cathepsin K polyclonal antibody (1:500, Abcam, ab19027) overnight at 4 °C, and subsequently with Alexa Fluor 647-conjugated donkey anti-goat IgG (A21082), or Alexa Fluor 647-conjugated donkey anti-rabbit IgG (A31573) (1:400, Invitrogen) for 3 h at room temperature. Sections were further incubated with DAPI (4′,6-diamidino-2-phenylindole, 5 μg/ml, Invitrogen D1306) to stain nuclei prior to imaging.

### RNAscope in situ hybridization

Samples were fixed in 4% paraformaldehyde overnight at 4 °C, and then cryoprotected. Frozen sections at 14 μm were prepared on positively charged glass slides. In situ hybridization was performed with RNAscope Multiplex Fluorescent Detection Kit v2 (Advanced Cell Diagnostics 323100) using an *Fgfr3* (440771) probe according to the manufacturer's protocol.

### Imaging

Images were captured by an automated inverted fluorescence microscope with a structured illumination system (Zeiss Axio Observer Z1 with ApoTome.2 system) and Zen 2 (blue edition) software. The filter settings used were: FL Filter Set 34 (Ex. 390/22, Em. 460/50 nm), Set 38 HE (Ex. 470/40, Em. 525/50 nm), Set 43 HE (Ex. 550/25, Em. 605/70 nm), Set 50 (Ex. 640/30, Em. 690/50 nm), and Set 63 HE (Ex. 572/25, Em. 629/62 nm). The objectives used were: Fluar 2.5x/0.12, EC Plan-Neofluar 5x/0.16, Plan-Apochromat 10x/0.45, EC Plan-Neofluar 20x/0.50, EC Plan-Neofluar 40x/0.75, and Plan-Apochromat 63x/1.40. Images were typically tile-scanned with a motorized stage, Z-stacked, and reconstructed by a maximum intensity projection (MIP) function. Differential interference contrast (DIC) was used for objectives higher than 10x. Representative images of at least three independent biological samples are shown in the figures. Quantification of cells on sections was performed using NIH Image J software.

### Cell dissociation

Soft tissues and distal epiphyseal growth plates were removed from dissected femurs and then cut off proximal ends.

For harvesting bone marrow stromal cells, the periosteum was carefully scraped off using a raspatory. Then, femurs were cut in half along with the long axis and incubated with 2 Wunsch units of Liberase TM (Sigma/Roche 5401127001) in 2 ml Ca²⁺, Mg²⁺-free Hank's Balanced Salt Solution (HBSS, Sigma H6648) at 37 °C for 60 min. on a shaking incubator (ThermomixerR, Eppendorf). After cell dissociation, cells were mechanically triturated using an 18-gauge needle with a 1-ml Luer-Lok syringe (BD) and a pestle with a mortar (Coors Tek), and subsequently filtered through a 70-μm cell strainer (BD) into a 50-ml tube on ice to prepare single cell suspension. These steps of mechanical trituration and filtration were repeated for five times, and dissociated cells were collected in the same tube.

For harvesting periosteal cells, the proximal and distal ends were capped by bone wax (SMI) and incubated with 2 Wunsch units of Liberase TM in 2 ml HBSS at 37 °C for 40 min. on a shaking incubator. After cell dissociation, cells were mechanically triturated using an 18-gauge

needle, and subsequently filtered through a 70-μm cell strainer (BD) into a 50-ml tube on ice to prepare a single-cell suspension.

For cell culture experiments, cells were resuspended in an appropriate amount of culture medium and counted on a hemocytometer. For flow cytometry analysis or cell sorting, cells were pelleted and resuspended in Flow Cytometry Staining Buffer (Invitrogen/eBioscience) (for flow cytometry) or in 1% BSA/PBS (for cell sorting) for subsequent purposes.

## Flow cytometry

Dissociated cells were incubated with the following antibodies (1:500, Invitrogen/eBioscience) in Flow Cytometry Staining Buffer on ice for 30 min. eFluor450-conjugated CD31 (390, endothelial/platelet, 48-0311-82), CD45 (30F-11, hematopoietic, 48-0451-82), Ter119 (TER119, erythrocytes, 48-5921-82), allophycocyanin (APC)-conjugated CD31 (390, endothelial/platelet, 17-0311-82), CD45 (30F-11, hematopoietic, 17-0451-82), Ter119 (TER119, erythrocytes, 17-5921-82), CD105 (MJ7/18, enoglin, 17-1051-82), phycoerythrin (PE)-conjugated CD51 (RMV-7, integrin alpha V, 12-0512-81), and PerCP-eFluor 710-conjugated CD200 (OX90, 46-5200-80). For CD90.2 staining, Alexa Fluor 700-conjugated CD90.2 (30-H12, Thy-1.2, Biolegend, 105319) were used.

Flow cytometry analysis was performed using a four-laser BD LSR Fortessa (Ex. 405/488/561/640 nm) and FACSDiva software. Acquired raw data were further analyzed on FlowJo software (TreeStar). Representative plots of at least three independent biological samples are shown in the figures.

## Single-cell RNA-seq analysis of FACS-isolated cells

Both male and female mice from the same litter were used to pool FACS-isolated tdTomato+ cells. Cell sorting was performed using a four-laser BD FACS Aria III (Ex.407/488/561/640 nm) high-speed cell sorter with a 100 μm nozzle. tdTomato+ cells were directly sorted into ice-cold DPBS/10% FBS, pelleted by centrifugation, and resuspended in 10 μl DPBS/1% FBS. Cell numbers were quantified by Countless II Automated Cell Counter (Thermo Fisher) before loading onto the Chromium Single Cell 3' v3.1 or Single Cell ATAC v2 microfluidics chips (10x Genomics Inc., Pleasanton, CA) following the manufacturer's recommended protocols. cDNA or gDNA libraries were sequenced by Illumina NovaSeq 6000. The sequencing data were first preprocessed using the 10X Genomics software Cell Ranger. For alignment purposes, we generated and used a custom genome fasta and index file by including the sequences of *tdTomato-WPRE* to the mouse genome (mm10).

## Single-cell sequencing analysis

To define the unique molecular features of the potential stem cell population, we employed our previously developed approach, called linked inference of genomic experimental relationships (LIGER)[29]. A typical LIGER analysis begins by applying standard preprocessing workflows to each modality independently. In this manuscript, we analyzed data measured by three different modalities: single-cell RNA sequencing (scRNA-seq), single-nucleus chromatin accessibility sequencing (snATAC-seq), and single-cell multiome ATAC + RNA sequencing. For all analyses in this manuscript, we used the default values for all key parameters. For selecting variable features in joint analyses, we always selected from gene expression data only and tuned the threshold to select approximately 3,000 key features. For clarity, we list the preprocessing steps for each data type below:

**Single-cell gene expression.** scRNA-seq data were analyzed using standard preprocessing pipelines in LIGER, including digital gene expression (DGE) matrix construction, normalization, feature selection, and scaling (but not centering).

**Single-nucleus chromatin accessibility sequencing.** We first transformed the snATAC-seq, a genome-wide epigenomic measurement, into gene-level counts by counting the total number of ATAC-seq reads within the gene body and promoter region (3 kilobases upstream) of the transcription start site in each cell. Then the data were processed with standard preprocessing pipelines in LIGER[51].

**10X Multiome ATAC + RNA sequencing.** The transcriptomic profiles were processed by typical preprocessing pipelines for scRNA-seq, and the epigenomic profiles were processed using the same pipeline for snATAC-seq.

After preprocessing, we performed cross-modality analyses following a typical LIGER workflow, including joint factorization and clustering by iNMF, clustering refining by Louvain community detection, and UMAP dimensional reduction using default parameters. All key steps, example results, and interpretations are described more in detail and formally in our previously published step-by-step protocol[51].

## RNA velocity analysis

We counted spliced and unspliced reads for each gene using velocyto (**velocyto**[30],) with default parameters for 10X Genomics data. This produced a loom file for each sample, which we loaded into scVelo (**scvelo**[31]) for subsequent analysis. Loom files were prepared by following the standard protocol using velocyto. Prior to velocity analysis, the bam files were sorted by position with samtools (**samtools**[52]) and the genome annotation (mm10_msk.gtf) was obtained from Ensembl. Using the aligned bam files, we quantified the unspliced and spliced abundances in loom format.

Next, we processed the loom file with typical scvelo preprocessing steps, including gene filtering using a threshold of 10 counts, normalization, variable gene selection (20 minimum shared counts, 5500 genes), log transformation, and nearest neighbor graph construction (K = 30 nearest neighbors). This step generates a velocity graph with a dimension of *n* rows and *n* columns (*n* indicates the total number of cells). Ultimately, we inferred RNA velocity using the likelihood-based dynamical model. Since various cell types contain differential kinetics that could not be well explained by a single model of the overall dynamics, we applied the differential kinetic test to adjust for the potential discrepancy among clusters.

## Illustration of differentiation potential by simplex plots

We developed a novel visualization method for illustrating the transcriptomic or epigenomic affinities between each cell type and three main classes of bone marrow stromal cells as ternary (or simplex) diagrams (**ggtern**[53]). This method begins by selecting the top differentially expressed features based on the data preprocessed above. We employed the Wilcoxon rank-sum test to select the top 30 differentially expressed genes (DEGs) or differentially accessible peaks (DAPs) from each cluster. Next, we quantified each cell's similarity to the three cell fates (chondrocyte, osteoblast, and reticular) by averaging the Euclidean distances between each cell and all the cells in each of the three terminal clusters. This step generates a similarity matrix with *n* rows (number of cells) and three columns. To make the similarities calculated with cells from diverse tissues comparable in the simplex space, we applied −log10 transformation and min-max scaling by feature to rescale the similarity matrix. To format the similarity matrix into three-dimensional simplex coordinates, we then re-normalized the coordinates by the cell to ensure the total of the three respective variables sum to unity (100%).

Note that the cell type labels for the scRNA-seq and snATAC-seq data shown in Fig. 2 were generated from a LIGER integration. In contrast, the single-cell multiome RNA + ATAC results shown in Fig. S2 are from experimentally measured cell correspondences.

## scVelo and simplex velocity

We adapted a previously developed method that inferred RNA velocity on scRNA-seq datasets (**velocyto**[30] and **scvelo**[31]), for projecting the cell fate potential of individual cells towards terminal cell fates onto simplex plots.

We used the RNA velocity results, calculated as described above, but projected the velocity vectors onto the 3-dimensional ternary coordinates calculated as described in the previous section. We defined the future differentiation potential as the averaged velocity graph edge weight between each cell and all the cells in each of the three apex clusters. The inferred potential was visualized using three arrows that point to each of the three vertices (each direction also indicated by a different color). The lengths of the arrows were weighted using the differentiation potential normalized within each cell. To make the arrows easier to see, we binned the simplex space into 40 grid locations, took the mean of the differentiation potential of all cells that fall into each location, and drew the arrows in the centroid of each.

## Accessibility peak calling and transcription factor binding site analysis

To enrich the most accessible regions in each cell type, we employed a previously developed method MACS2 (**macs2**[54]) for generating signal tracks and peak calling. We first split the raw snATAC-seq fragment file by clusters using the labels generated by LIGER in previous sections. Next, we run MACS2 on fragment files with parameters settings adapted from a previously published paper (**snapATAC**[55]) with the effective genome size "-g" set to mouse genome "mm". We then reapplied the preprocessing pipelines for snATAC-seq data in previous sections to transform these newly generated peaks into gene-level counts (cell-by-peak matrix), and performed cross-modality integrations with scRNA-seq data.

We investigated the most abundant transcription factor binding sites in each cluster with the function "findMotifsGenome.pl" from a computational package Homer (**homer**[56]). To begin with, we first employed the Wilcoxon rank-sum test to identify the most differentially accessible peaks (DAPs) within each cluster using the cell-by-peak matrix generated in the previous step. Next, we filtered these DAPs to retain only those with adjusted $p$ value <0.05, ordered by log fold-change, picked the top 1000 DAPs, and transformed into BED files. We then performed HOMER analysis with the parameter settings provided by a previously published paper (**snapATAC**[55]).

## Single-cell copy number variation analysis

We investigated the tumor characteristics of the aberrant chondrocyte-like cells with computational packages that infer genome-wide copy number variations, such as CopyKAT[43] and InferCNV (https://github.com/broadinstitute/inferCNV). We first adopted CopyKAT to identify aneuploid cells based on scRNA-seq data preprocessed by LIGER. Since the original package only supports GRCh38 (v28) as the reference genome, we modified the source code to add support for mm10 to enable analyses of our mouse data. To achieve this, we re-estimated the copy number resolution using the pipelines described in the original CopyKAT paper, including ordering all genes by their genomic positions, defining gene intervals and finding the median, computing gene detection rate, and calculating the minimum genome bin size. We then performed the detection analysis with Spearman correlation using default parameters.

We also employed InferCNV to investigate copy number variation with the labels of aneuploid and normal cells generated by CopyKAT. To provide genomic positions for each gene, we downloaded the BED file for all gene entries in mm10 from the UCSC Genome Browser and ordered all genes by their genomic positions. We also set the cut-off value for the minimum average read counts per gene among reference cells to be 0.1 since our data was generated by the 10X Genomics platform. Then, we ran inferCNV on the preprocessed scRNA-seq data using default parameters.

## Gene ontology enrichment analysis

To perform Gene Ontology (GO) term enrichment analysis, we first performed the Wilcoxon rank-sum test on scRNA-seq data to select top differentially expressed genes (DEGs) in Δp53 and control cells within the aberrant chondrocyte-like cells. We then filtered these marker genes to retain only those with adjusted $p$ value <0.05 and log fold-change >1. Next, we utilized the web interface of GOrilla (**gorilla**[57]) to perform GO analysis. For parameter settings, we chose: (1) organism: "Mus musculus"; (2) running mode: "Two unranked lists of genes", with DGEs from Δp53 and control cells as the target set separately and all genes in the scRNA-seq dataset as the background set; (3) ontology: "Process"; (4) check the box "Show output also in REVIGO". For the "$P$ value threshold", we used 0.3 for control cells and 0.5 for Δp53 cells due to the substantial number of GO terms selected from Δp53 cells.

We then performed REVIGO (**revigo**[58]) analysis on the exported list of GO terms using default parameters. In the results page, we exported the outputs by pressing the button "Export to CSV". We then customized the REVIGO R code for plotting to modify the color scale and text size.

## Whole-exome sequencing (WES)

The bone with osteosarcoma-like lesions (Osteosarcoma) and the bone on the contralateral side without macroscopically apparent lesions (Contralateral no osteosarcoma-like lesion) obtained from *Fgfr3-creER*; *Trp53*$^{fl/fl}$; *R26R*$^{tdTomato}$ mice at 9 months of age (pulsed at P21) were dissected and fixed in 4% PFA for overnight and then kept in PBS. The phenotypically normal bone obtained from *Trp53*$^{fl/fl}$; *R26R*$^{tdTomato}$ littermate mice at 9 months of age were used as the background control (Control). Genomic DNA was purified using QIAamp DNA Micro Kit (QIAGEN 56304). Sample quality control was performed using the 5400 Fragment Analyzer system (Agilent). OD260/280 = 1.8–2.0, no contamination, no degradation samples were used for subsequent analyses. The genomic DNA was randomly sheared into short fragments with a size of 180–280 bp. The obtained fragments were end-repaired, A-tailed, and further ligated with Illumina adapters. The fragments with adapters were PCR amplified, size selected, and purified. Hybridization capture of libraries was proceeded according to the following procedures. Briefly, the prepared libraries were hybridized in the buffer with biotin-labeled probes, and magnetic beads with streptavidin were used to capture the exons of genes. Subsequently, non-hybridized fragments were washed out and probes were digested. The captured libraries were enriched by PCR amplification. The library was checked with Qubit and real-time PCR for quantification and a bioanalyzer for size distribution detection. Quantified libraries will be pooled and sequenced on Illumina platforms with PE150, according to effective library concentration and data amount required.

To detect copy number variations for WES, we utilized the following three bam files: Fgfr3-p53 cKO femoral osteosarcoma tumor, Fgfr3-p53 cKO contralateral femur without osteosarcoma-like lesion, and *Trp53*$^{fl/fl}$; *R26R*$^{tdTomato}$ normal femur as the background control. The processed bam files were obtained after processing steps, which include (1) sorting and indexing with SAMtools[59] and (2) marking duplicate reads with Picard (http://broadinstitute.github.io/picard/). The coverage and depth were computed based on the processed files. WES analysis steps were carried out as stated in the ExomeCopy documentation[60]. In brief, we set the target bed file as the Agilent mouse mm9 exon kit and the reference genome as mm9. Then, we calculated the GC content, squared-GC, and the genomic width, generated background using WES of *Trp53*$^{fl/fl}$; *R26R*$^{tdTomato}$ bone, and log-transformed the background counts. Ultimately, we inferred the copy number states by modeling the read counts of the sample of

interest while using GC content, squared-GC, and range width as covariates. For genome-wide CNV visualization, we used regioneR, karyoploteR, and CopyNumberPlots[61,62] (https://github.com/bernatgel/CopyNumberPlots). We annotated genes by employing R packages, including TxDb.Mmusculus.UCSC.mm9.knownGene, Mus.musculus, and GenomicRanges (https://doi.org/10.18129/B9.bioc.TxDb.Mmusculus.UCSC.mm9.knownGene, https://doi.org/10.18129/B9.bioc.Mus.musculus)[63].

## Bulk RNA-seq analysis of cultured cells

Cultured Fgfr3^CE-tdTomato^+ (WT, isolated at P23, Passage 8) and Fgfr3^CE-Δp53 tdTomato^+ (cKO, isolated at P28 and 4M, Passage 6–8) clones (all pulsed at P21) were collected into ice-cold DPBS/10% FBS and pelleted by centrifugation. Total RNA was isolated using PicoPure RNA Isolation Kit (KIT0204, Thermo Fisher), followed by a DNA-free DNA removal kit (AM1906, Thermo Fisher) to remove contaminating genomic DNA. RNA Integrity Number (RIN) was assessed by Agilent 2100 Bioanalyzer RNA 6000 Pico Kit. Samples with RIN >8.0 were used for subsequent analyses. Complementary DNAs were prepared by SMART-Seq v4 Ultra Low Input RNA Kit for Sequencing (Clontech 634888). Post-amplification quality control was performed by Agilent TapeStation DNA High Sensitivity D1000 Screen Tape system. DNA libraries were prepared by Nextera XT DNA Library Preparation Kit (Illumina) and submitted for NextGen sequencing (Illumina NovaSeq 6000). DNA libraries were sequenced using the following conditions; six samples per lane, 50 cycle single-end read. Reads files were downloaded and concatenated into a single .fastq file for each sample. The quality of the raw reads data for each sample was checked using FastQC to identify quality problems. Tuxedo Suite software package was subsequently used for alignment (using TopHat and Bowtie2), differential expression analysis, and post-analysis diagnostics. FastQC was used for a second round of quality control (post-alignment). HTSeq/DESeq2 was used for differential expression analysis using UCSC mm10 as the reference genome sequence. Only non-ambiguously mapped reads were counted. Meta-analysis of RNA-seq data was performed using iPathwayGuide software (Advaita).

## Colony-forming assay and trilineage differentiation

Nucleated bone marrow cells were plated into tissue culture six-well plates (BD Falcon) at a density of <10^5 cells/cm^2 and cultured in low-glucose DMEM with GlutaMAX supplement (Gibco 10567022) and 10% mesenchymal stem cell-qualified FBS (Gibco 12662029) containing penicillin-streptomycin (Sigma P0781) for 10–14 days. Cell cultures were maintained at 37 °C in a 5% CO₂ incubator. Representative images of tdTomato^+ colonies and following methylene blue staining are shown in the figures. Colonies marked by tdTomato were individually subcloned. A tile-scanned virtual reference of tdTomato^+ colonies and cloning cylinders (Bel-Art) were used to isolate an individual colony. Single-cell-derived clones of tdTomato^+ cells were cultured in a basal medium. Cells at Passages 4–8 were used for trilineage differentiation. To induce chondrocyte differentiation, cells were transferred into a non-tissue culture-coated V-bottom 96 well plate (Corning 3896). The plate was centrifuged at 150×*g* for 5 min at room temperature to pellet the cells, and the supernatant was carefully aspirated. StemPro Chondrogenesis medium (Thermo Fisher A1007101) was gently added, and the plate was centrifuged at 150×*g* for 5 min at room temperature to pellet the cells. The pellet was cultured in the differentiation medium with exchanges into fresh media every 3–4 days for up to 3 weeks, each time with centrifugation at 150×*g* for 5 min at room temperature to repellet the cells. Cell pellets were fixed with 70% ethanol for 5 min. and stained for Alcian Blue Staining Solution (Millipore TMS-010-C) for 30 min. To induce osteoblast differentiation, cells were plated on a 48-well plate and cultured with αMEM with GlutaMAX supplement (Gibco 32571036) with 10% FBS containing 1 μM dexamethasone (Sigma D4902), 10 mM β-glycerophosphate (Sigma G9422) and 50 μg/ml

ascorbic acid (Sigma A5960) with exchanges into fresh media every 3–4 days for up to 3 weeks. Cells were fixed with 4% paraformaldehyde for 5 min and stained for 2% Alizarin Red S (Sigma A5533) for 30 min. To induce adipocyte differentiation, cells were plated on a 48-well plate and cultured with αMEM with GlutaMAX supplement with 10% FBS containing 1 μM dexamethasone (Sigma D4902), 0.5 mM IBMX (3-Isobutyl-1-methylxanthine, Sigma I5879), and 1 μg/ml insulin (Sigma I6634) with exchanges into fresh media every 3–4 days for up to 2 weeks. Cells were stained with LipidTOX Green (Invitrogen H34475, 1:200 in basal medium) for 3 h at 37 °C, or fixed with 4% paraformaldehyde for 5 min and stained for Oil Red O (Sigma O0625).

## Quantitative reverse-transcription PCR (qRT-PCR)

Cultured Fgfr3^CE-tdTomato^+ clones (WT, isolated at P23 after being pulsed at P21, over Passage 8) were collected into ice-cold DPBS/10% FBS and pelleted by centrifugation. Total RNA was isolated using PicoPure RNA Isolation Kit (Thermo Fisher KIT0204), followed by a DNA-free DNA removal kit (Thermo Fisher AM1906) to remove contaminating genomic DNA. Complementary DNA was prepared using SuperScript IV First-Strand Synthesis System (Thermo Fisher 18091050). The expression was quantified by qRT-PCR using CFX96 Touch Real-Time PCR Detection System (Bio-Rad) and TaqMan Universal PCR Master Mix (Thermo Fisher 43044437) with probes of trilineage-specific marker genes, including *Acan* (Mm00545794_m1), *Col2a1* (Mm01309565_m1), *Dmp1* (Mm01208363_m1), *Sp7* (Mm04209856_m1), *Pparg* (Mm00440940_m1), and *Adipoq* (Mm04933656_m1). The relative amount of each mRNA was normalized to *Gapdh* (Mm99999915_g1) expression.

## Intrafemoral transplantation of cultured cells

Six to eight weeks old NSG mice were used as recipients for cell transplantation. Mice were anesthetized through a nose cone, in which 1.5–2% isoflurane is constantly provided with oxygen through a vaporizer. For intrafemoral transplantation, bilateral femurs were operated. After the skin incision, the cruciate ligaments of the knee were carefully separated using a dental excavator, and a hole was made in the intercondylar region of the femurs using a 26-1/2-gauge needle (BD309597). Subsequently, $1 \times 10^5$ cells/10 μl of cultured Fgfr3^CE-tdTomato^+ (WT, isolated at P23, Passage 8) or Fgfr3^CE-Δp53 tdTomato^+ (cKO, isolated at P28 and 4M, Passage 6–8) clonal cells (all pulsed at P21) were transplanted into the marrow space using a 26-1/2-gauge needle and 1-ml syringe. The hole was filled with bone wax (Med vet international), and the incision line was sutured.

## Three-dimensional microcomputed tomography analysis

Specimens were placed in a 19-mm diameter specimen holder and scanned using a microCT system (μCT100 Scanco Medical, Bassersdorf, Switzerland). Scan settings were: voxel size 12 μm, 70 kVp, 114 μA, 0.5-mm AL filter, and integration time 500 ms. Analysis was performed using the manufacturer's software (Scanco μCT Evaluation Program). Trabecular analysis was performed in a region beginning 120 μm below the growth plate and ending after 360 μm (30 slices); a fixed global threshold of 18% (180 on a grayscale of 0–1000) was used to segment mineralized tissue from mineralized tissue. Cortical analysis was performed in the region beginning 600 μm below the growth plate and ending after 360 μm (30 slices) using a fixed global threshold of 23% (230 on a grayscale of 0–1000); a secondary ROI was used to exclude any trabecular bone located in the region of cortical analysis.

## Statistical analysis

Results are presented as mean values ± SD. Statistical evaluation was conducted by the two-tailed Mann–Whitney's *U*-test, two-tailed Wilcoxon matched-pairs signed-rank test, two-tailed one-way ANOVA followed by Dunnett's multiple comparison test, or Log-rank

(Mantel–Cox) test using GraphPad Prism software. A *P* value of <0.05 was considered significant. No statistical method was used to pre-determine the sample size. The sample size was determined on the basis of previous literature and our previous experience to give sufficient standard deviations of the mean so as not to miss a biologically important difference between groups. The experiments were not randomized. All of the available mice of the desired genotypes were used for experiments. The investigators were not blinded during experiments and outcome assessments. One femur from each mouse was arbitrarily chosen for histological analysis. Genotypes were not particularly highlighted during quantification.

### Reporting summary

Further information on research design is available in the Nature Portfolio Reporting Summary linked to this article.

### Data availability

Source data are provided with this paper. The raw data generated during and/or analyzed during the current study are available from the corresponding author on reasonable request. The single-cell RNA-seq, single-cell ATAC-seq, and bulk RNA-seq data presented herein have been deposited in the National Center for Biotechnology Information (NCBI)'s Gene Expression Omnibus (GEO), and are accessible through the GEO Series accession number, SuperSeries GSE185794, including GSE185785-GSE185793 and GSM5623768-GSM5623790. Reference genome mm10 (Ensembl release 93) for transcript annotation is publicly available on Ensembl, including a gtf file [http://ftp.ensembl.org/pub/release-93/gtf/mus_musculus/Mus_musculus.GRCm38.93.gtf.gz] and a fasta file [http://ftp.ensembl.org/pub/release-93/fasta/mus_musculus/dna/Mus_musculus.GRCm38.dna_sm.primary_assembly.fa.gz]. Source data are provided with this paper.

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

## Acknowledgements

This research was supported by grants from the National Institute of Health (R01DE026666 and R01DE030630 to N.O., R01HG010883 to J.D.W., and R01DE029181 to W.O.) and JSPS grant JP21H03124 and JST FOREST Program JPMJFR2111 Japan to Y.M. We thank H.M. Kronenberg of Massachusetts General Hospital for *Osx-creER* mice, T. Nagasawa of Osaka university for *Cxcl12-GFP* mice, K. Saiya-Cork of the University of Michigan Flow Cytometry Core, G. Gavrilina and W. Fillipak of the University of Michigan Transgenic Animal Model Core, D. Sheltraw, J. Opp of the University of Michigan Advanced Genomics Core, R. Tagett of the University of Michigan Bioinformatics Core, and N. Sakagami for supporting this study.

## Author contributions

Y.M., J.D.W., and N.O. conceived the project. Y.M., J.L., J.D.W., and N.O. wrote the manuscript. Y.M., J.L., A.K.Y.C., C.T.A., M.N., Y.A., N.S., J.D.W., and N.O. performed the experiment. T.L.S. generated the mice. W.O. and K.Y. critiqued and critically revised the manuscript.

## Competing interests

The authors declare no competing interests.
