## [Peer Review File · Nature Communications]

Bone marrow endosteal stem cells dictate active osteogenesis
and aggressive tumorigenesisEditorial Note: This manuscript has been previously reviewed at another journal that is not operating a transparent peer review scheme. This document only contains reviewer comments and rebuttal letters for versions considered at *Nature Communications*.

REVIEWERS' COMMENTS

Reviewer #1 (Remarks to the Author):

The authors have responded well to most of initial critiques and the revised manuscript has been significantly improved. This manuscript also well addressed the existence of stromal cells with osteoblast-chondrocyte transitional (OCT) identities in the bone marrow. We would appreciate that the authors included new transplantation experiments showing the Osteoblast-chondrocyte transitional identities of Fgfr3+ endosteal stromal cells. Further, authors included a new FGFR3-GFP mouse model and revealed that at least a fraction of ALPL+Fgfr3-GFP+ cells co-expressed ACAN in the endosteal space.

However, the only concern that I have is the TRANSIENT nature of these OCT stem cells in young mice. This is the same question from reviewer 2 regarding the definition of “transitional” stem cells. Given authors did well to define that there is no stem cell function of Fgfr3^{CE-9M} stromal cells, it would be very important to clarify that the OCT stem cells are “transient” or “transitional” stem cells in the title as well as in abstract.

I also recommend minor modifications that there are a few spots missed using italics for *in vivo* and *ex vivo*.

Line 79 and 345 *in vivo* needs to be in italics

Line 342 *ex vivo* needs to be in italics

Reviewer #2 (Remarks to the Author):

In this revision to the manuscript, Matsushita et al. have, for the most part, answered the questions raised in the previous reviews well, and the manuscript is strengthened for it. However, in many cases (not only in response to the questions that I raised but also in response to those of the other reviewers) the authors have not incorporated their answers (in the form of both text and figures) into the revised manuscript. While I am of course aware that papers can quickly balloon to unwieldy sizes and this ought to be avoided wherever possible, by inclusion only of pertinent information, in this case it seems that the authors have left out many points that are as likely to be raised by readers as by the reviewers. I hope the authors can find ways to address this, e.g. by expedient use of language in the main text and with SI figures, such that important points of discussion are not omitted from the manuscript. Otherwise, what is the point of the review? Specific examples from the questions I raised are given below. However, as noted, I think the same issues arise for many of the questions raised by other reviewers also.

1. The answer to Q1 on the naming of “OCT” is not incorporated into the manuscript. Some (abbreviated) form of this discussion is important to guide the reader in the first results subsection.

2. In response to Q2, Figs R2-2-1 and R2-2-2 are not included in the revised ms. Why not include these figs in SI? Or at the very least mention in the m/s that these experiments have been done to study robustness and the necessity for focused sets of DEGs/DEPs? The results of which are likely to be of interest to readers other than me.

3. No incorporation of the discussion of sex differences (Q4) into the revised manuscript. This is not acceptable.

4. No incorporation of response to Q6 into the manuscript. This is optional, and the authors can argue for its exclusion if they so choose, however I think it does constitute an interesting and perhaps important point.

5. In response to Q8, the authors produced Fig R2-8-1, which I suggested might be easier to interpret than Fig 2b. Indeed, I think Fig R2-8-1 is much clearer than Fig 2b vis a vis the location of Fgfr3+ and Gas1+ cells. Of course this is a subjective matter, and if the authors strongly disagree with the preference for visualization, then I will respect this. However, in my opinion, the version in Fig R2-8-1 presents the data in an immediately more comparable way via the UMAP background.

Reviewer #3 (Remarks to the Author):

The authors have provided additional functional experiments that demonstrate more specifically the contributions of endosteal Fgfr3+ cells.

Reviewer #4 (Remarks to the Author):

The authors addressed the comments very adequately by performing several additional experiments and adapting the text accordingly. The stem cell properties of the identified FGFR3+ endosteal cells are now proven by the transplantation studies followed by bone repair, self-renewal and secondary CFU.

It remains however unclear why the authors call these endosteal stem cells 'osteoblast-chondrocyte transitional cells', while they describe them as 'positioned in the middle of the three major cell types' (line 100) and they express molecular markers from chondrocytes, osteoblasts and reticular cells (line 102-103). In addition, these cells have trilineage differentiation potential. The term OCT is somewhat misleading as it suggests that these cells are derived from osteoblasts and are becoming chondrocytes, which is not proven in this study. The authors provide data that some of the FGFR3+ cells express both osteoblast (ALPL) and chondrocyte (ACAN) markers, which might be a characteristic of bipotential cells. Of note, it has been shown that chondrocytes express ALPL (Ishikawa Y, 1987). It is therefore advised to change the terminology 'OCT' as it gives the impression that this cell type is 'in transition' between osteoblasts to chondrocytes, which is not proven.

Response to Reviewers and Editors: NCOMMS-22-47516-T

RE: *Bone marrow endosteal stem cells dictate active osteogenesis and aggressive tumorigenesis*

We are extremely delighted to hear that our manuscript is provisionally accepted for *Nature Communications*. Please see below for our point-by-point response to the reviewers' comments, as well as the editors' requests.

Response to Reviewer #1

<Reviewer>

The authors have responded well to most of initial critiques and the revised manuscript has been significantly improved. This manuscript also well addressed the existence of stromal cells with osteoblast-chondrocyte transitional (OCT) identities in the bone marrow. We would appreciate that the authors included new transplantation experiments showing the Osteoblast-chondrocyte transitional identities of Fgfr3+ endosteal stromal cells. Further, authors included a new FGFR3-GFP mouse model and revealed that at least a fraction of ALPL+Fgfr3-GFP+ cells co-expressed ACAN in the endosteal space.

However, the only concern that I have is the TRANSIENT nature of these OCT stem cells in young mice. This is the same question from reviewer 2 regarding the definition of "transitional" stem cells. Given authors did well to define that there is no stem cell function of Fgfr3^{CE-9M} stromal cells, it would be very important to clarify that the OCT stem cells are "transient" or "transitional" stem cells in the title as well as in abstract.

<Response>

We appreciate your comments. As per the editor's request, we maintained our original title. However, we have emphasized the transitional nature of Fgfr3+ OCT stem cells in the abstract as well as in the introduction, as demonstrated below.

Abstract: These Fgfr3-creER-marked endosteal stromal cells contribute to a stem cell fraction in young stages, which is later replaced by Lepr-cre-marked stromal cells in adult stages.

Introduction: These Fgfr3⁺ stem/stromal cells with OCT identities are abundant in young bone marrow and depleted in old bone marrow, denoting their transitional nature. ... The term "transitional" emphasizes the unique feature of these cells that are particularly abundant in young bone marrow.

<Reviewer>

I also recommend minor modifications that there are a few spots missed using italics for in vivo and ex vivo.

Line 79 and 345 in vivo needs to be in italics

Line 342 *ex vivo* needs to be in italics

<Response>

Changed as recommended, thank you very much.

Response to Reviewer #2

<Reviewer>

In this revision to the manuscript, Matsushita et al. have, for the most part, answered the questions raised in the previous reviews well, and the manuscript is strengthened for it. However, in many cases (not only in response to the questions that I raised but also in response to those of the other reviewers) the authors have not incorporated their answers (in the form of both text and figures) in to the revised manuscript. While I am of course aware that papers can quickly balloon to unwieldy sizes and this ought to be avoided wherever possible, by inclusion only of pertinent information, in this case it seems that the authors have left out many points that are as likely to be raised by readers as by the reviewers. I hope the authors can find ways to address this, e.g. by expedient use of language in the main text and with SI figures, such that important points of discussion are not omitted from the manuscript. Otherwise, what is the point of the review? Specific examples from the questions I raised are given below. However, as noted, I think the same issues arise for many of the questions raised by other reviewers also.

<Response>

Thank you very much for these important comments. We have now added **all the data in the rebuttal letter to the revised manuscript**, and restructured the main figures and the supplementary figures as appropriate associated with explanations in the main text. We now have 10 main figures and 9 supplementary figures.

<Reviewer>

1. The answer to Q1 on the naming of “OCT” is not incorporated into the manuscript. Some(abbreviated) form of this discussion is important to guide the reader in the first results subsection.

<Response>

We have now added the following sentences in the introduction and in the first results subsection.

Introduction: These Fgfr3⁺ stem/stromal cells with OCT identities are abundant in young bone marrow and depleted in old bone marrow, denoting their transitional nature. Of note, we define OCT identities as a state with some characteristics of both osteoblasts and chondrocytes, instead of cell type plasticity between osteoblasts and chondrocytes. The term “transitional” emphasizes the unique feature of these cells that are particularly abundant in young bone marrow.

Results (Line 110-114): In the following section, we define OCT identities as a state with some characteristics of both osteoblasts and chondrocytes, but not as a state in which cells are transitioning between osteoblasts and chondrocytes. The term “transitional” emphasizes the unique feature of these cells that are abundant in young bone marrow but depleted in old bone marrow.

<Reviewer>

2. In response to Q2, Figs R2-2-1 and R2-2-2 are not included in the revised ms. Why not include these figs in SI? Or at the very least mention in the m/s that these experiments have been done to study robustness and the necessity for focused sets of DEGs/DEPs? The results of which are likely to be of interest to readers other than me.

<Response>

These figures (Figs. R2-2-1 and R2-2-2) have been incorporated as **Supplementary Figure 3** in the revised manuscript, accompanied with the following sentence in the Results:

Results (Line 142-147): We selected the top 30 differentially expressed genes (DEGs) or differentially accessible peaks (DAPs) from each cluster; the ternary plots showed a very similar trend using 500 or 1,000 differentially expressed features (Supplementary Fig. 3a). Including the whole cell type as the reference (instead of Osteoblast 1, Chondrocyte 1 and Reticular 1) distorted the shape of each plot, demonstrating the negative effect of defining the cell fates using the less pure and more transitional clusters (Supplementary Fig. 3b).

<Reviewer>

3. No incorporation of the discussion of sex differences (Q4) into the revised manuscript. This is not acceptable.

<Response>

This figure (Fig. R2-4-1) has been incorporated as Supplementary Figure 2d in the revised manuscript, accompanied with the following sentence in the Results:

Results (Line 107-109): Computational inference of the sex of each cell revealed that the female and male cells maintained a rather consistent 3:1 ratio across all clusters at P21 (Supplementary Fig. 1e), suggesting that the key clusters we identified were not sex-dependent, and sex differences did not confound our analyses.

<Reviewer>

4. No incorporation of response to Q6 into the manuscript. This is optional, and the authors can argue for its exclusion if they so choose, however I think it does constitute an interesting and perhaps important point.

<Response>

Following the reviewer's suggestion, we have now added the following sentence in the Results.

Results (Line 544-547): Interestingly, $Fgfr3^{CE-WT}$ clones were more variable than $Fgfr3^{CE-\Delta p53}$ clones on PC2 (Fig. 9d), indicating that $Fgfr3^+$ SSC clones may undergo phenotypic convergence due to p53 loss and carry similar molecular characteristics.

<Reviewer>

5. In response to Q8, the authors produced Fig R2-8-1, which I suggested might be easier to interpret than Fig 2b. Indeed, I think Fig R2-8-1 is much clearer than Fig 2b vis a vis the location of Fgfr3+ and Gas1+ cells. Of course this is a subjective matter, and if the authors strongly disagree with the preference for visualization, then I will respect this. However, in my opinion, the version in Fig R2-8-1 presents the data in an immediately more comparable way via the UMAP background.

<Response>

Following the reviewer's suggestion, we have now incorporated Fig. R2-8-1 as the main **Figure 2d** in the revised manuscript.

Response to Reviewer #3

<Reviewer>

The authors have provided additional functional experiments that demonstrate more specifically the contributions of endosteal Fgfr3+ cells.

<Response>

Thank you very much for your positive assessment on our revised manuscript.

Response to Reviewer #4

<Reviewer>

The authors addressed the comments very adequately by performing several additional experiments and adapting the text accordingly. The stem cell properties of the identified FGFR3+ endosteal cells are now proven by the transplantation studies followed by bone repair, self-renewal and secondary CFU. It remains however unclear why the authors call these endosteal stem cells 'osteoblast-chondrocyte transitional cells', while they describe them as 'positioned in the middle of the three major cell types' (line 100) and they express molecular markers from chondrocytes, osteoblasts and reticular cells (line 102-103). In addition, these cells have trilineage differentiation potential. The term OCT is somewhat misleading as it suggests that these cells are derived from osteoblasts and are becoming chondrocytes, which is not proven in this study. The authors provide data that some of the FGFR3+ cells express both osteoblast (ALPL) and chondrocyte (ACAN) markers, which might be a characteristic of bipotential cells. Of note, it has been shown that chondrocytes express ALPL (Ishikawa Y, 1987). It is therefore advised to change the terminology 'OCT' as it gives the impression that this cell type is 'in transition' between osteoblasts to chondrocytes, which is not proven.

<Response>

Thank you very much for your important comments. As per the editor's comments, we decided to maintain the OCT terminology. However, we have clarified the definition of OCT in the revised manuscript, in the introduction, results and discussion, as demonstrated below.

Introduction: Of note, we define OCT identities as a state with some characteristics of both osteoblasts and chondrocytes, instead of cell type plasticity between osteoblasts and chondrocytes. The term "transitional" emphasizes the unique feature of these cells that are particularly abundant in young bone marrow.

Results: In the following section, we define OCT identities as a state with some characteristics of both osteoblasts and chondrocytes, but not as a state in which cells are transitioning between osteoblasts and chondrocytes. The term "transitional" emphasizes the unique feature of these cells that are abundant in young bone marrow but depleted in old bone marrow.

Discussion: We emphasize that our definition of OCT identities does not infer cell type plasticity between two differentiated cell types, wherein cells are transitioning between osteoblasts and chondrocytes. We define OCT identities as a state with some characteristics of both osteoblasts and chondrocytes. Our findings that these OCT cells that are particularly abundant in young bone marrow and depleted in aged bone marrow denote their transitional nature.